# Expression of WIPI2B counteracts age-related decline in autophagosome biogenesis in neurons

Andrea KH Stavoe[1], Pallavi P Gopal[1†], Andrea Gubas[2], Sharon A Tooze[2], Erika LF Holzbaur[1]*

[1]Department of Physiology, Perelman School of Medicine, University of Pennsylvania, Philadelphia, United States; [2]Molecular Cell Biology of Autophagy Laboratory, The Francis Crick Institute, London, United Kingdom

**Abstract** Autophagy defects are implicated in multiple late-onset neurodegenerative diseases including Amyotrophic Lateral Sclerosis (ALS) and Alzheimer's, Huntington's, and Parkinson's diseases. Since aging is the most common shared risk factor in neurodegeneration, we assessed rates of autophagy in mammalian neurons during aging. We identified a significant decrease in the rate of constitutive autophagosome biogenesis during aging and observed pronounced morphological defects in autophagosomes in neurons from aged mice. While early stages of autophagosome formation were unaffected, we detected the frequent production of stalled LC3B-negative isolation membranes in neurons from aged mice. These stalled structures recruited the majority of the autophagy machinery, but failed to develop into LC3B-positive autophagosomes. Importantly, ectopically expressing WIPI2B effectively restored autophagosome biogenesis in aged neurons. This rescue is dependent on the phosphorylation state of WIPI2B at the isolation membrane, suggesting a novel therapeutic target in age-associated neurodegeneration.
DOI: https://doi.org/10.7554/eLife.44219.001

*For correspondence:
holzbaur@pennmedicine.upenn.edu

Present address: †Department of Pathology, Yale University School of Medicine, New Haven, United States

Competing interests: The authors declare that no competing interests exist.

## Introduction

Aging is a complex process that often impairs physiological and tissue function. Further, age is the most relevant risk factor for many prominent diseases and disorders, including cancers and neurodegenerative diseases (*Niccoli and Partridge, 2012*). Macroautophagy (hereafter referred to as autophagy) is an evolutionarily conserved, cytoprotective degradative process in which a double membrane engulfs intracellular cargo for breakdown and recycling (*Cuervo et al., 2005*; *Rubinsztein et al., 2011*). The autophagy pathway has been directly implicated in aging in model organisms (*Cuervo, 2008*; *Rubinsztein et al., 2011*).

Neurons are post-mitotic, terminally differentiated cells that must maintain function in distal compartments throughout the lifetime of a human. These maintenance mechanisms may wane as a person ages, potentially contributing to neuronal dysfunction and death. Accordingly, misregulation of autophagy has been associated with multiple age-related neurodegenerative diseases, including Alzheimer's disease (AD), Parkinson's disease, Huntington's disease, and amyotrophic lateral sclerosis (ALS) (*Menzies et al., 2017*; *Nixon, 2013*; *Yamamoto and Yue, 2014*). Furthermore, specifically disrupting autophagy in neurons results in neurodegeneration in animal models (*Hara et al., 2006*; *Komatsu et al., 2006*; *Zhao et al., 2013*).

Despite the implication of this pathway in neurodegenerative disease, autophagy is best understood for its roles in maintaining cellular homeostasis in yeast and mammalian cells in response to acute stressors such as starvation (*Abada and Elazar, 2014*; *Hale et al., 2013*; *Mariño et al., 2011*; *Reggiori and Klionsky, 2013*; *Son et al., 2012*; *Wu et al., 2013*). Much less is known about how

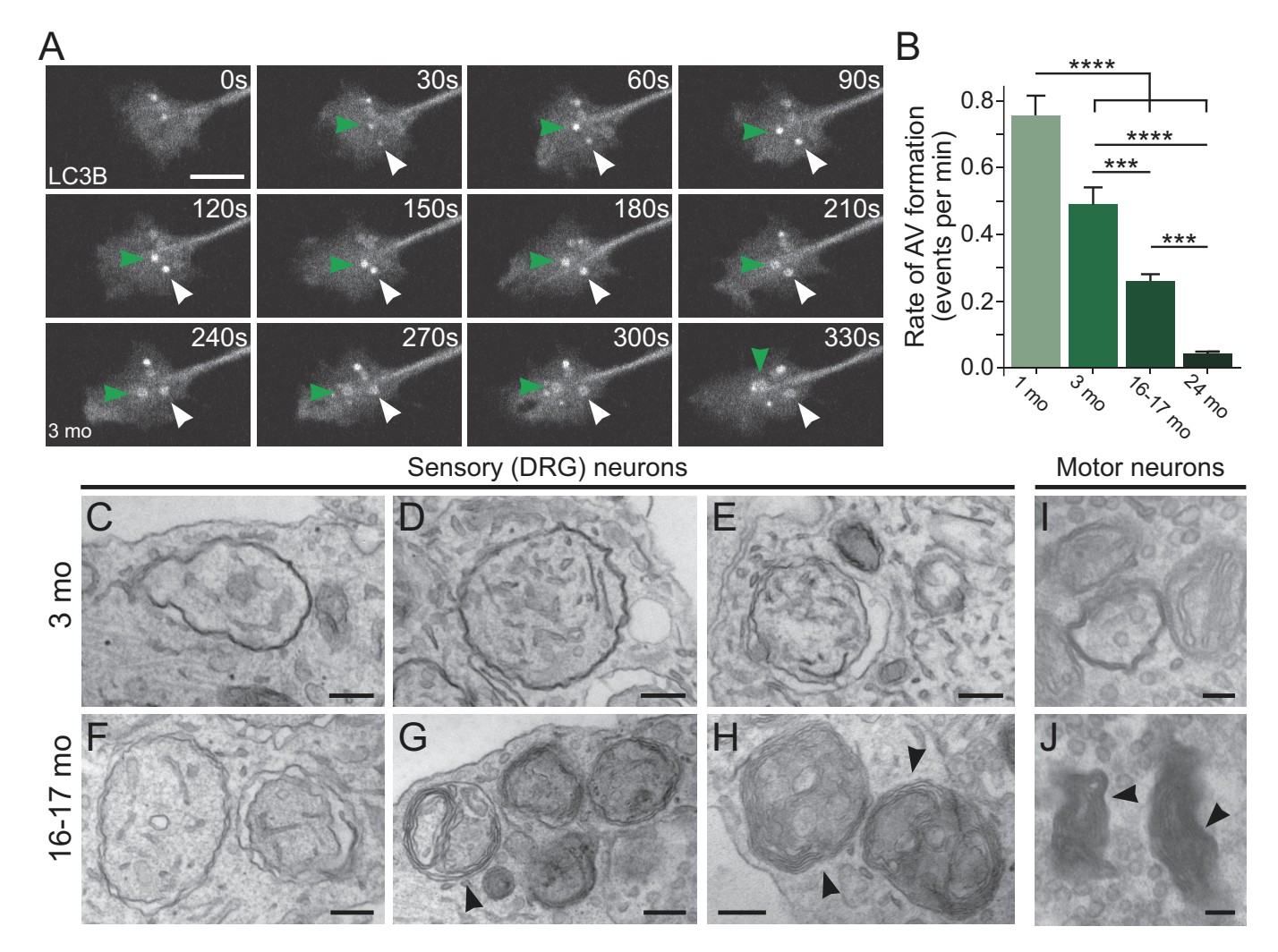

**Figure 1.** Autophagosome biogenesis decreases with age and results in aberrant AV formation in mammalian neurons. (A) Time series of GFP-LC3B in the distal axon of a DRG neuron from a young adult mouse. Green and white arrowheads each follow one autophagosome biogenesis event. Retrograde is to the right. Scale bar, 2 μm. (B) Quantification of the rate of autophagic vesicle (AV) biogenesis (assayed by GFP-LC3B puncta formation per minute) in DRG neurons from young (one mo, light green), young adult (three mo, green), aged (16–17 mo, dark green), and advanced aged (24 mo, very dark green) mice (mean ± SEM; n ≥ 54 neurons from three biological replicates). ***p<0.0005; ****p<0.0001 by one-way ANOVA test with Tukey's multiple comparisons test. (C–E) Representative electron micrographs of autophagosomes in DRG distal tips from young adult mice. AVs are composed of a continuous double membrane enclosing engulfed cytoplasm. Scale bars, 200 nm. (F–H) Representative electron micrographs of autophagosomes in DRG distal tips from aged mice. AVs contain multiple, ruffled double membranes (G, H). Scale bars, 200 nm. (I–J) Electron micrographs of autophagosomes in the presynaptic compartment of neuromuscular junctions (NMJs) from young adult (I) and aged (J) mice. Scale bars, 100 nm. Arrowheads indicate multilamellar membranes in DRGs and NMJs.

DOI: https://doi.org/10.7554/eLife.44219.003

The following figure supplement is available for figure 1:

**Figure supplement 1.** Morphological differences in autophagosomes in neurons from young adult and aged mice.

DOI: https://doi.org/10.7554/eLife.44219.004

autophagy is regulated in neurons. Robust, constitutive autophagy functions at a constant, basal level in neurons both in vitro and in vivo. Autophagosomes are generated distally at the axon terminal or synapse and are then actively transported back to the soma during maturation to a fully-acidified degradative compartment (*Fu et al., 2014*; *Hara et al., 2006*; *Hollenbeck, 1993*; *Komatsu et al., 2007*; *Maday et al., 2012*; *Neisch et al., 2017*; *Soukup et al., 2016*; *Stavoe et al., 2016*; *Yang et al., 2013*; *Yue et al., 2009*). In contrast to the pronounced induction of autophagy in

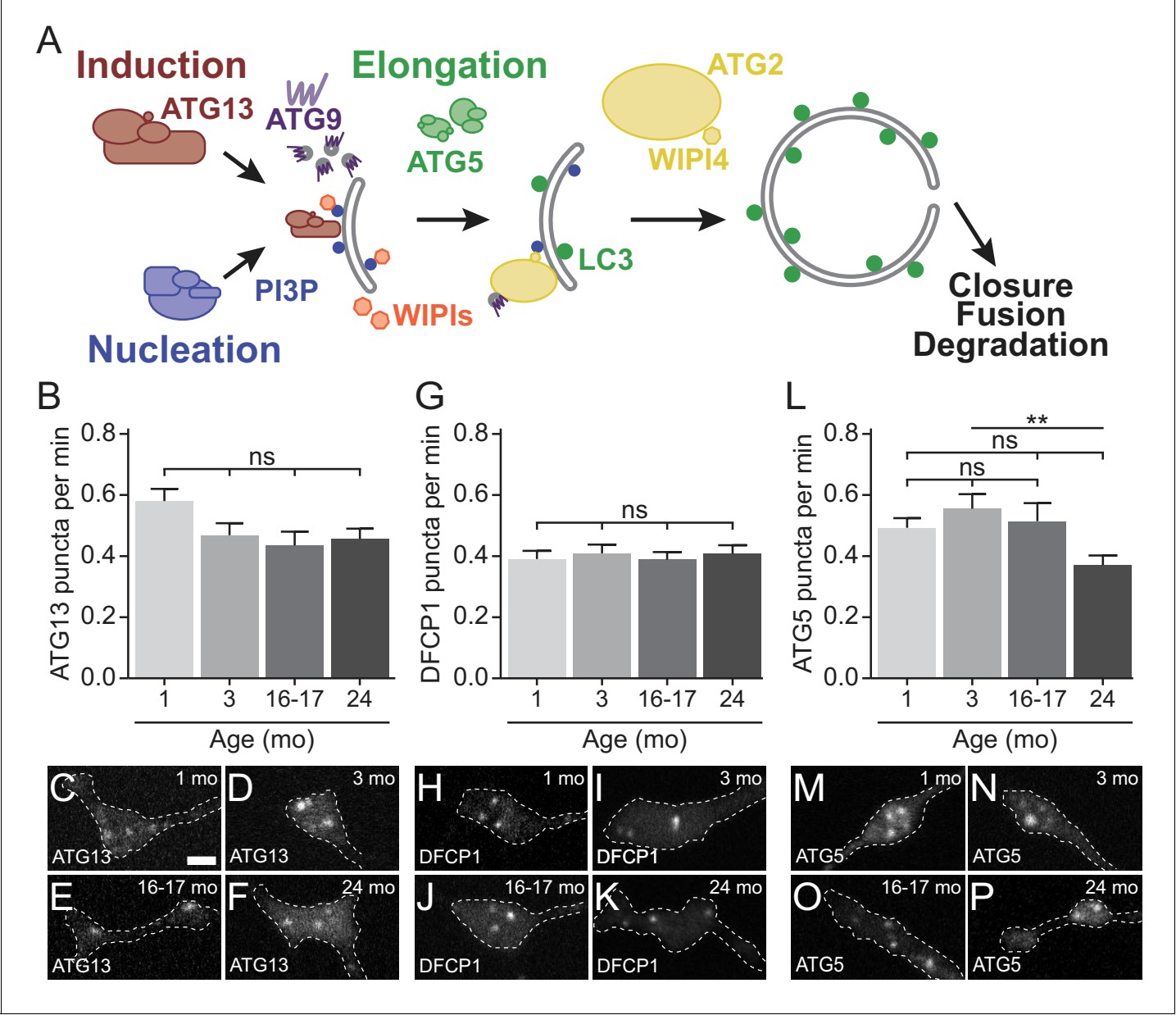

**Figure 2.** Early autophagosome biogenesis components do not change with age. (**A**) Schematic of autophagy pathway, focusing on the protein complexes involved in autophagosome biogenesis: induction (red), nucleation (blue), elongation (green), and ATG2/WIPI4 (yellow). ATG9, a multi-pass transmembrane protein is in purple. The product of the nucleation complex, PI3P, is depicted as a blue dot, while LC3-II, the product of the elongation complex, is depicted as a green dot. WIPI1 and WIPI2, which bind to PI3P, are displayed in orange. (**B**) Quantification of the rate of mCh-ATG13 puncta formation in live-cell imaging of DRG neurons from young, young adult, aged, and advanced aged mice (mean ± SEM; n ≥ 28 neurons from three biological replicates). ns, not significant by Kruskal-Wallis ANOVA test with Dunn's multiple comparisons test. (**C–F**) Representative micrographs of mCh-ATG13 in the distal tip of DRG neurons from young (**C**), young adult (**D**), aged (**E**), and advanced aged (**F**) mice. (**G**) Quantification of the rate of Halo-DFCP1 puncta in DRG neurons from young, young adult, aged, and advanced aged mice (mean ± SEM; n ≥ 18 neurons from three biological replicates). ns, not significant by Kruskal-Wallis ANOVA test with Dunn's multiple comparisons test. (**H–K**) Representative micrographs of Halo-DFCP1 in the distal tip of DRG neurons from young (**H**), young adult (**I**), aged (**J**), and advanced aged (**K**) mice. (**L**) Quantification of the rate of mCh-ATG5 puncta in DRG neurons from young, young adult, aged, and advanced aged mice (mean ± SEM; n ≥ 34 neurons from three biological replicates). ns, not significant; **p<0.001 by Kruskal-Wallis ANOVA test with Dunn's multiple comparisons test. (**M–P**) Representative micrographs of mCh-ATG5 in the distal tip of DRG neurons from young (**M**), young adult (**N**), aged (**O**), and advanced aged (**P**) mice. Scale bar in C, 2 μm, for C-F, H-K, M-P.
DOI: https://doi.org/10.7554/eLife.44219.005

other systems by cellular stressors, there is little evidence that neuronal autophagy is substantially upregulated by either proteomic stress (*Maday et al., 2012*; *Wong and Holzbaur, 2014*) or nutrient deprivation (*Maday and Holzbaur, 2016*). While recent progress has firmly linked autophagy and aging (*Chang et al., 2017*; *Hansen et al., 2018*), little is known about how this essential homeostatic mechanism in neurons is affected by aging.

Autophagosome biogenesis, conserved from yeast to humans, involves over 30 proteins that act in distinct protein complexes to engulf either bulk cytoplasm or specific cargo within a double-membrane. A signature autophagy protein, LC3B, is used to label autophagosomes, as it is processed to a lipidated form that becomes tightly associated with the limiting membrane of the developing autophagosome. We have previously used live imaging to examine autophagosome biogenesis in dorsal root ganglion (DRG) neurons from transgenic mice expressing GFP-LC3B (*Maday et al., 2012*; *Maday and Holzbaur, 2014*). Importantly, the spatially-specific pathway for constitutive axonal autophagy that has been extensively characterized in DRG neurons (*Fu et al., 2014*; *Maday et al., 2012*; *Maday and Holzbaur, 2014*; *Wong and Holzbaur, 2014*) has been confirmed across multiple models, including hippocampal and cortical neurons in vitro (*Lee et al., 2011*; *Maday and Holzbaur, 2014*) and motor, touch, and interneurons in vivo in *Drosophila* and *C. elegans* (*Chang et al., 2017*; *Neisch et al., 2017*; *Soukup et al., 2016*; *Stavoe et al., 2016*). However, unlike hippocampal or cortical neurons, which are typically isolated from embryonic or early postnatal rodents, DRG neurons can be isolated from mice of any age and grow robustly in culture following dissection. As rates of autophagosome biogenesis in DRG neurons model those seen in vivo (*Soukup et al., 2016*; *Stavoe et al., 2016*) and longitudinal studies indicate biogenesis rates remain constant over time in neurons in vitro (*Maday and Holzbaur, 2014*), DRGs represent a powerful model system to investigate autophagosome formation in mammalian neurons from aged mice with high temporal and spatial resolution.

Young neurons appear to clear dysfunctional organelles and protein aggregates very efficiently (*Boland et al., 2008*), but few studies have examined autophagy in aged neurons. Since age is the most relevant shared risk factor in neurodegenerative disease (*Niccoli and Partridge, 2012*), elucidating how autophagy changes in neurons with age is crucial to understanding neurodegenerative diseases.

Here, we examine how autophagy is altered with age in primary neurons from mice. We find that the rate of autophagosome biogenesis decreases in neurons with age. This decrease is not due to a change in the kinetics of either initiation or nucleation during autophagosome formation. Instead, we find that the majority of autophagosome biogenesis events in neurons from aged mice exhibit pronounced stalling, remaining ATG13-positive and failing to recruit lipidated LC3B with normal kinetics. We observe pronounced morphological differences in autophagic vesicles in neurons from aged mice, including an increased frequency of multilamellar membranes, similar to observations of neurons from the brains of aging Alzheimer's patients (*Nixon et al., 2005*). Importantly, depletion of WIPI2 in neurons from young adult mice was sufficient to decrease the rate of autophagosome biogenesis to that of aged mice, while overexpression of WIPI2B in neurons from aged mice was sufficient to return the rate of autophagosome biogenesis to that found in neurons from young adult mice. Further, we find that the rescue of autophagosome biogenesis depends on the phosphorylation state of WIPI2B at the isolation membrane, suggesting that dynamic phosphorylation of WIPI2B regulates autophagosome biogenesis. Thus, while the rate of autophagosome biogenesis decreases in aged neurons, this decrease can be rescued by the restoration of a single autophagy component, suggesting a novel therapeutic target for future studies.

## Results

### Autophagosome biogenesis at the axon terminal decreases with age

Since impaired autophagy has been implicated in the pathogenesis of neurodegeneration and age is the most relevant risk factor for neurodegenerative disease, we used the GFP-LC3B probe to assess how biogenesis rates change with age in primary DRG neurons dissected from mice of four different ages: 1-month-old young mice, 3-month-old young adult mice, 16–17 month-old aged mice, and 24-month-old advanced aged mice. We produced robust cultures of DRG neurons harvested from mice aged from 1 to 24 months; neurons harvested from all ages extended long neurites, and we did not

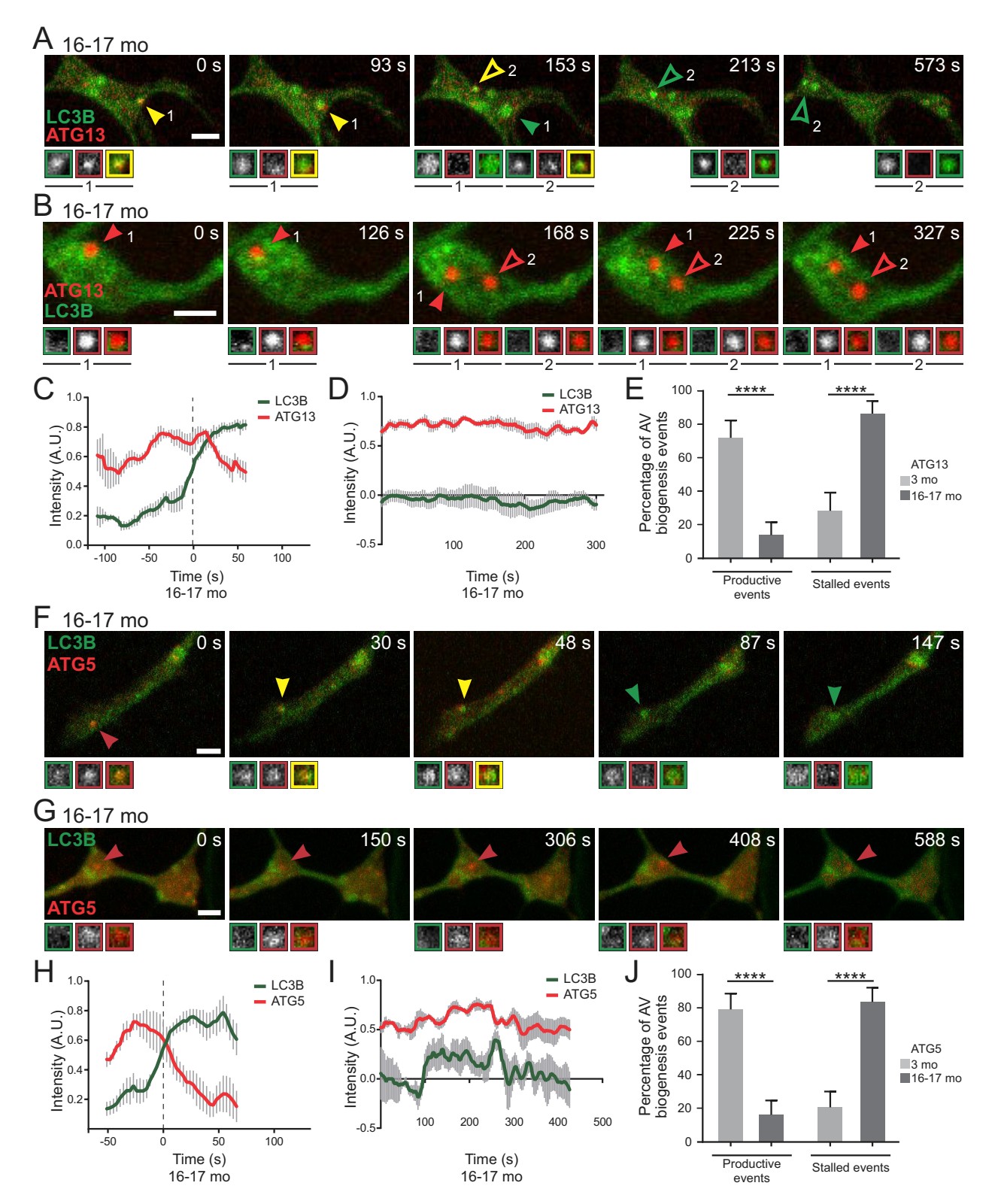

**Figure 3.** Stalled AVs predominate in neurons from aged mice. (A–B) Time series of merge micrographs of mCh-ATG13 and GFP-LC3B from live cell imaging of the distal neurite of DRGs from aged mice depicting a productive (A) or a stalled (B) autophagosome biogenesis event. Yellow arrowheads denote colocalization of mCh-ATG13 and GFP-LC3B; green arrowheads denote a GFP-LC3B-positive punctum from which mCh-ATG13 has dissociated; red arrowheads denote mCh-ATG13-positive puncta that fail to recruit GFP-LC3B; solid arrowheads track one punctum, hollow arrowheads follow a

*Figure 3 continued on next page*

*Figure 3 continued*

second punctum. Magnified views of denoted puncta are shown below full micrograph; border color represents channel or colocalization state in merge. Retrograde is to the right. Scale bars, 2 μm. (C–D) Individual intensity profiles were averaged to improve signal-to-noise of mCh-ATG13 (red) and GFP-LC3B (green) for productive (C) and stalled (D) AVs (mean ± SEM; n ≥ 5 biogenesis events from five neurons from three biological replicates). (E) Quantification of the proportion of total mCh-ATG13-positive AV biogenesis events (both productive and stalled events) in DRG neurons from young adult (light gray) and aged (dark gray) mice (mean ± 95% confidence interval; n ≥ 62 AVs from three biological replicates for each condition). ****p<0.0001 by Fisher's exact test. (F–G) Time series of merge micrographs of mCh-ATG5 and GFP-LC3B in the distal neurite of DRGs from aged mice depicting a productive (F) or stalled (G) autophagosome biogenesis event. Arrowheads point to puncta magnified below micrograph; colors denote channel or colocalization state in merge. Scale bars, 2 μm. (H–J) Mean intensity profiles of mCh-ATG5 (red) and GFP-LC3B (green) for productive (H) and stalled (I) AVs (mean ± SEM; n = 5 productive biogenesis events from five neurons or n = 4 stalled biogenesis events from four neurons from three biological replicates). Vertical dashed line in (C and H) indicates the half-maximum of GFP-LC3B intensity, which was used to align the traces. (J) Quantification of the proportion of total mCh-ATG5-positive AV biogenesis events (both productive and stalled events) in DRG neurons from young adult and aged mice (mean ± 95% confidence interval; n ≥ 62 AVs from three biological replicates for each condition). ****p<0.0001 by Fisher's exact test. See also *Videos 1–6*.

DOI: https://doi.org/10.7554/eLife.44219.012

The following figure supplement is available for figure 3:

**Figure supplement 1.** Productive and stalled AVs occur in the same axonal tips.

DOI: https://doi.org/10.7554/eLife.44219.013

detect significant loss of viability with age (data not shown). We used live-cell spinning disk fluorescence microscopy to examine autophagosome biogenesis at the axon tips of DRG neurons in culture with high spatial and temporal specificity.

We identified autophagosome biogenesis events as the formation of discrete GFP-LC3B puncta visible above the background cytoplasmic GFP-LC3B signal (*Figure 1A*). These puncta enlarged over approximately three minutes to form a 1 μm autophagosome. Strikingly, we found that the rate of autophagosome biogenesis significantly decreased with age, corresponding to a 53% decrease in autophagosome biogenesis in aged neurons compared to neurons from young adult mice. Furthermore, the decrease was even more pronounced in neurons from advanced aged mice (*Figure 1B*). These data indicate that the rate of autophagosome biogenesis, as detected by the generation of GFP-LC3B-positive puncta, decreases in axon terminals with increasing age.

In subsequent experiments, we focused on 16–17 month-old aged mice, given the significant decrease in the rate of autophagosome biogenesis observed at this time point relative to young adult mice and the relevance of this time point to the age of onset for age-associated neurodegenerative diseases such as ALS and AD.

## Morphological differences are common in neuronal autophagosomes of aged mice

To further characterize changes in autophagic vesicle (AV) biogenesis in neurons during aging, we used transmission electron microscopy to compare the ultrastructure of AVs at axon terminals of neurons from young adult and aged mice. We observed stereotypical double-membrane structures with heterogeneous contents in the axonal tips of neurons from young adult mice (*Figure 1C–E*, *Figure 1—figure supplement 1A and C–E*). However, in neurons from aged mice we more frequently observed aberrant AVs with a multilamellar (onion skin-like) structure (*Figure 1F–H*, *Figure 1—figure supplement 1B and F–M*). Quantitative analysis indicated that only 34.0% of AVs (n = 153 AVs) observed in the distal tips of neurons from 16 to 17 month-old mice were morphologically normal, significantly different than the 80.4% of AVs judged to be morphologically normal in the axon tips of neurons from young adult mice (n = 56 AVs; p<0.0001 by unpaired two-tailed Fisher's exact test). The aberrant morphology of AVs observed in aged mice suggested misregulated membrane extension during AV biogenesis, consistent with previous observations that failure to lipidate LC3 at isolation membranes prevented closure and inhibited the degradation of the inner autophagosome membrane (*Tsuboyama et al., 2016*). Furthermore, these aberrant AVs were reminiscent of AVs previously observed in aged rodents (*Majeed, 1993*; *Majeed, 1992*) and in cortical biopsy specimens from patients with Alzheimer's disease (*Nixon et al., 2005*).

We next queried whether we could detect these age-related morphological differences in intact neuronal tissues, focusing on the prominent synapses that form between motor neurons and muscle

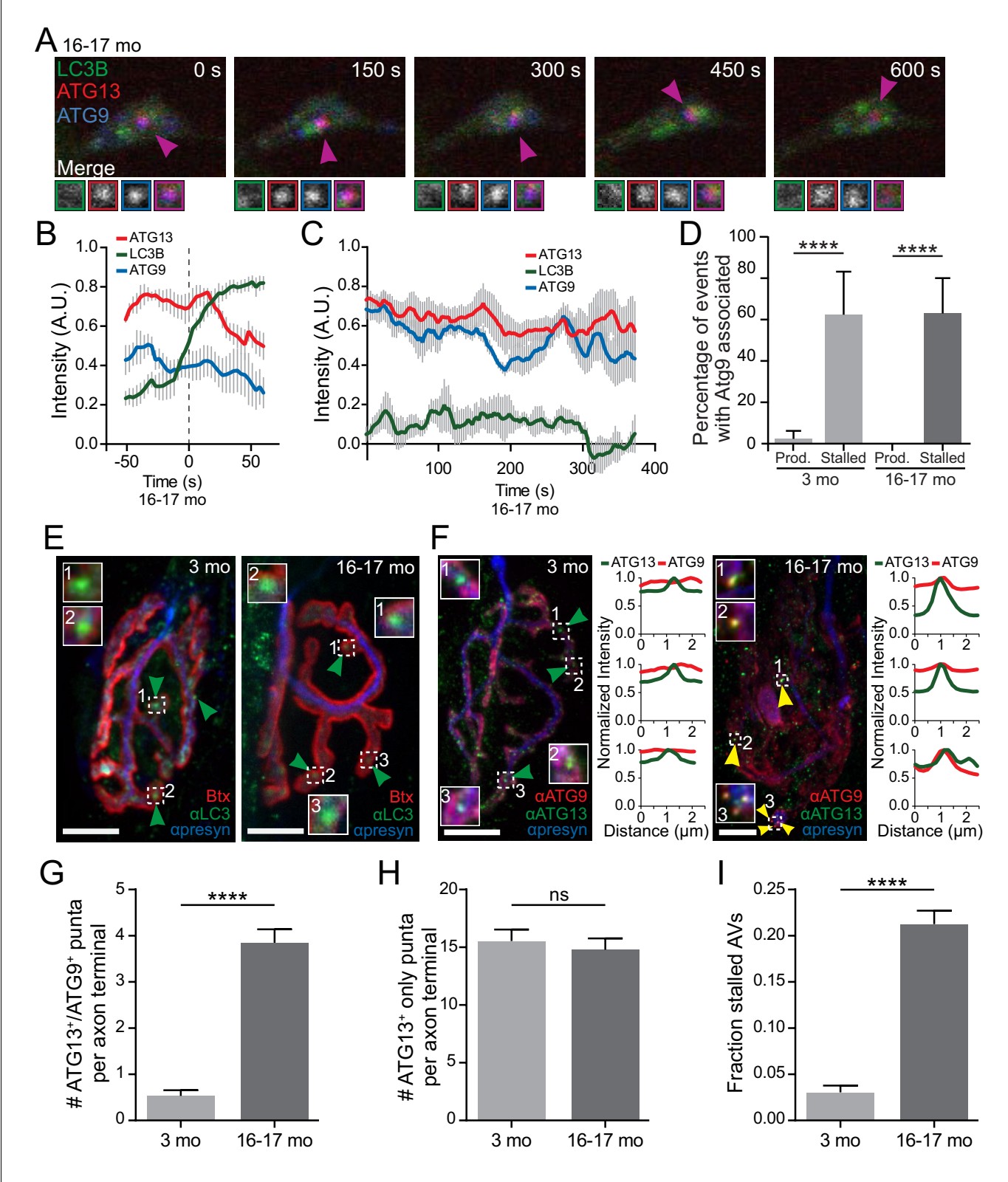

**Figure 4.** Atg9, a multi-pass transmembrane protein, aberrantly associates with stalled AVs in vitro and in vivo. (**A**) Time series of live imaging of mCh-ATG13, SNAP-ATG9, and GFP-LC3B in the distal neurite of a DRG neuron from an aged mouse depicting a stalled AV. Magenta arrowheads indicate colocalization between mCh-ATG13 and SNAP-ATG9 without GFP-LC3B. Magnified views of denoted puncta are shown below full micrograph; border color represents channel or colocalization state in merge. Retrograde is to the right. Scale bar, 2 μm. (**B–C**) Mean intensity profiles of mCh-ATG13 (red),

*Figure 4 continued on next page*

*Figure 4 continued*

SNAP-ATG9 (blue), and GFP-LC3B (green) for productive (**B**) and stalled (**C**) AVs in DRG distal tips from aged mice (mean ± SEM; n = 5 biogenesis events from five neurons from three biological replicates for each graph). Vertical dashed line in (**B**) indicates the half-maximum of GFP-LC3B intensity, which was used to align the traces. (**D**) Quantification of the percentage of AVs that have SNAP-ATG9 associated in the distal neurites of DRGs from young adult and aged mice (mean ± 95% confidence interval; n ≥ 17 for each age group). ****p<0.0001 by two-tailed Fisher's exact test. (**E–F**) Maximal projection micrographs of NMJs from young adult (three mo) and aged mice (16–17 mo). In panel (**E**), NMJs were stained with α-Bungarotoxin-tetramethylrhodamine (Btx) to stain the endplate, anti-SV2 together with anti-neurofilament H (both in blue) to visualize the presynaptic motor neuron, and anti-LC3B to visualize AVs at the synapse. In panel (**F**), NMJs from young and aged mice were stained with the presynaptic markers SV2 and neurofilmament H (together in blue), as well as antibodies to ATG9 (red) and ATG13 (green). Dashed boxes indicate magnified insets; line scans of ATG9 (red) and ATG13 (green) intensities at the indicated puncta are also shown. Scale bars, 10 μm. Arrowheads denote LC3B AVs (**E**) or the colocalization state of ATG13 with ATG9 at AVs (**F**). (**G–I**) Quantification of micrographs of NMJs from young adult (gray) and aged (dark gray) mice stained with Btx, anti-SV2, anti-neurofilament H, anti-ATG13, and anti-ATG9. (**G**) Quantification of stalled AVs, defined as colocalization of ATG13 and ATG9 at puncta, in the NMJ motor axon terminal. (**H**) Quantification of ATG13 puncta that do not have colocalized ATG9 in the NMJ motor axon terminal. (**I**) Quantification of the fraction of stalled AVs out of the total ATG13-positive puncta in the NMJ motor axon terminal. In G-I, mean ± SEM; n ≥ 62 motor axon terminals for each age from three biological replicates. ****p<0.0001, ns = 0.3443 by Mann-Whitney t tests. See also *Videos 3,4,7*.
DOI: https://doi.org/10.7554/eLife.44219.015

The following figure supplement is available for figure 4:

**Figure supplement 1.** NMJ immunohistochemistry.
DOI: https://doi.org/10.7554/eLife.44219.016

at the neuromuscular junction (NMJ). We used NMJs from young adult and aged mice to assess any age-related changes in autophagosomes in vivo. Again, we observed stereotypical double-membrane structures with heterogeneous contents in neurons from young mice (*Figure 1I*, *Figure 1—figure supplement 1O*). As we observed in DRG neurons in culture, we identified multilamellar structures in NMJs from aged mice in vivo (*Figure 1J*, *Figure 1—figure supplement 1N and P*).

## Pronounced stalling of autophagosome biogenesis is observed in aged neurons

Autophagosome biogenesis can be divided into stages: initiation/induction, nucleation, elongation, and membrane closure (*Figure 2A*). The initiation complex, including ATG13 and ULK1/ATG1, induces autophagosome biogenesis by phosphorylating other autophagy components (*Feng et al., 2014*; *Kamada et al., 2000*; *Reggiori et al., 2004*). The nucleation complex, including VPS34 and ATG14, generates phosphatidylinositol 3-phosphate (PI3P) at the site of autophagosome biogenesis (*Kihara et al., 2001*; *Obara et al., 2006*). Subsequently, the elongation complex, composed of two conjugation complexes, including ATG5, ATG12, and ATG16L1, is required to conjugate phosphatidylethanolamine (PE) to LC3 to yield LC3-II (*Tanida et al., 2004*). LC3-II is recruited to autophagosomes as the isolation membrane elongates during biogenesis and remains associated with autophagosomes until degradation of the internalized components. ATG9, a six-pass transmembrane protein, is thought to shuttle to the growing membrane with donor membrane (*Koyama-Honda et al., 2013*; *Orsi et al., 2012*; *Sekito et al., 2009*; *Suzuki et al., 2015*; *Yamamoto et al., 2012*; *Young et al., 2006*). The ATG2 and WIPI4 complex is thought to work in concert with ATG9 to tether and provide lipids to the growing membrane (*Chowdhury et al., 2018*; *Gómez-Sánchez et al., 2018*; *Osawa et al., 2019*; *Valverde et al., 2019*; *Wang et al., 2001*). Finally, the limiting membrane closes and fuses with itself to generate the unique double-membrane organelle. The autophagosome then undergoes retrograde transport along microtubules and subsequent fusion with lysosomes to degrade engulfed contents (*Figure 2A*) (*Xie and Klionsky, 2007*).

The observed decrease in the rate of autophagosome biogenesis that we measured by monitoring GFP-LC3B-positive puncta could result from alterations to the initiation, nucleation, or elongation complexes. To determine which stage of autophagosome biogenesis is affected by age, we used live-cell imaging to compare the kinetics of each step of the pathway (*Figure 2A*). We monitored the recruitment of the initiation complex by quantifying the appearance of fluorescent mCherry(mCh)-ATG13 puncta and observed similar kinetics of mCh-ATG13 recruitment in neurons from young, young adult, aged, and advanced aged mice (*Figure 2B–F*). To examine the kinetics of nucleation, we examined the recruitment of Double FYVE-containing protein 1 (DFCP1), which binds to PI3P, the product of the autophagy nucleation complex (*Figure 2A*). We did not detect a change

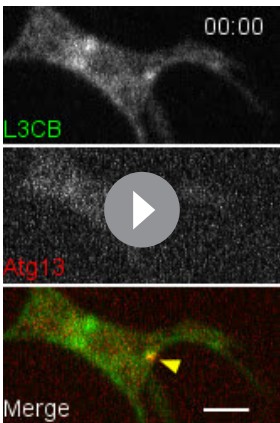

**Video 1.** Productive biogenesis events in a neuron from an aged mouse. GFP-LC3B and mCh-ATG13 in the distal neurite of a DRG neuron from an aged mouse depicting a productive autophagosome biogenesis event. In the merge movie (bottom), yellow arrowheads denote colocalization of ATG13 and LC3B; green arrowheads denote a LC3B-positive punctum from which ATG13 has dissociated; solid arrowheads track one punctum, hollow arrowheads follow a second punctum. Retrograde is to the right. Scale bar, 2 µm. Playback at five frames per second. Movie stills are shown in *Figure 3A*.

DOI: https://doi.org/10.7554/eLife.44219.006

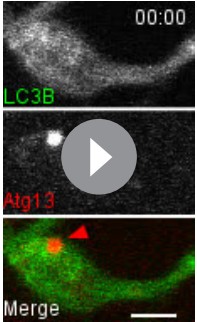

**Video 2.** Stalled biogenesis events in a neuron from an aged mouse. GFP-LC3B and mCh-ATG13 in the distal neurite of a DRG neuron from an aged mouse depicting a stalled autophagosome biogenesis event. In the merge movie (bottom), red arrowheads denote lack of colocalization between ATG13 and LC3B; solid arrowheads track one punctum, hollow arrowheads follow a second punctum. Retrograde is to the right. Scale bar, 2 µm. Playback at five frames per second. Movie stills are shown in *Figure 3B*.

DOI: https://doi.org/10.7554/eLife.44219.007

in DFCP1 puncta formation with age (*Figure 2G–K*). We examined elongation with the marker mCh-ATG5, and again observed no change between neurons from young, young adult, and aged mice; we did observe a decrease in the rate of mCh-ATG5 puncta in neurons from advanced aged mice compared to young adult mice (*Figure 2L–P*). Together, these data demonstrate that the decrease in the rate of autophagosome formation with age is not due to an alteration in the kinetics of the early stages of biogenesis.

Next we used dual-color live cell imaging to compare assembly dynamics in neurons from young adult and aged mice co-expressing GFP-LC3B and initiation component mCh-ATG13. We observed 'productive' biogenesis events in neurons from aged mice (*Figure 3A*, *Video 1*), very similar to those previously described in neurons from young adult (4–6 month-old) mice (*Maday and Holzbaur, 2014*). Quantitative analysis of the change in fluorescence intensity over time (*Figure 3C*) indicated that mCh-ATG13 transiently localizes to these puncta for 100 to 150 s; subsequent recruitment of GFP-LC3B to a mCh-ATG13-positive punctum coincided with a loss in mCh-ATG13 signal intensity.

Frequently, however, dual labeling of autophagosome biogenesis in neurons from aged mice revealed 'stalled' events, in which mCh-ATG13 puncta formed and were stably maintained for at least 5 min of a 10 min video; we observed that these stalled events also failed to recruit GFP-LC3B within the imaging window (*Figure 3B and D* and *Video 2*). In neurons from young adult mice, greater than 75% of observed events were productive AVs (*Figure 3E*). In striking contrast, we found that stalled events dominated in neurons from aged mice, representing greater than 75% of total events in aged neurons (*Figure 3E*).

We observed a similar distinction between productive and stalled events when we compared the recruitment kinetics of GFP-LC3B with elongation complex component mCh-ATG5 (*Figure 3F–G*; *Videos 3,4*). In neurons from aged mice, we observed stereotypical AV kinetics, in which the transient recruitment of mCh-ATG5 over approximately 100 s is followed by a steady increase in GFP-LC3B intensity (*Figure 3H*), similar to our observations with mCh-ATG13. Again, productive events predominated (>70% of total events) in neurons from young adult mice, while stalled events predominated (~80% of total events) in neurons from aged mice (*Figure 3I–J*). While these stalled events did not go on to produce GFP-LC3B-positive autophagosomes, stalled AVs remained dynamic within the confines of the axon tip rather than remaining tethered in place (*Figure 3B and G*, *Video 2,4*). Furthermore, both stalled and productive AVs could be found within the same axonal tip

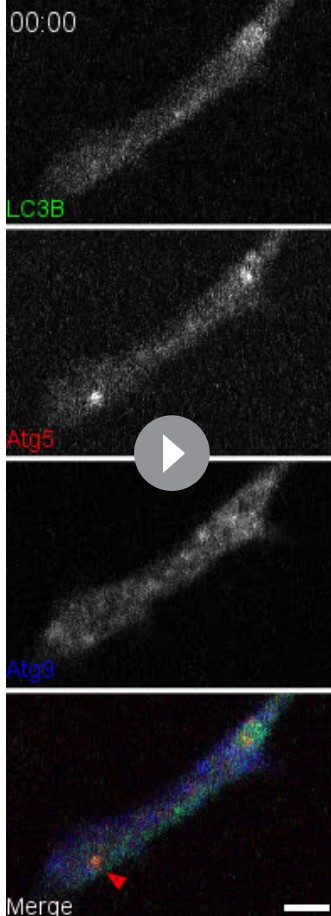

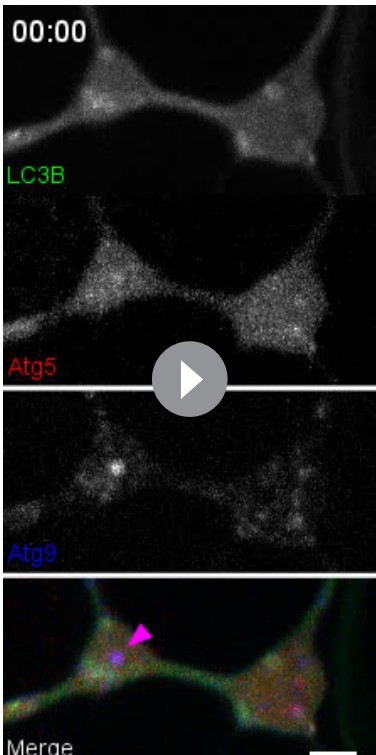

**Video 4.** Stalled biogenesis events in a neuron from an aged mouse. GFP-LC3B, mCh-ATG5, and SNAP-ATG9 in the distal neurite of a DRG neuron from an aged mouse depicting a stalled autophagosome biogenesis event. In the merge movie (bottom), magenta arrowheads denote colocalization between ATG5 and ATG9 without LC3B. Retrograde is to the right. Scale bar, 2 μm. Playback at five frames per second. Movie stills are shown in *Figure 3G*.

DOI: https://doi.org/10.7554/eLife.44219.009

**Video 3.** Productive biogenesis events in a neuron from an aged mouse. GFP-LC3B, mCh-ATG5, and SNAP-ATG9 in the distal neurite of a DRG neuron from an aged mouse depicting a productive autophagosome biogenesis event. In the merge movie (bottom), yellow arrowheads denote colocalization of ATG5 and LC3B without ATG9; green arrowheads denote a LC3B-positive punctum from which ATG5 has dissociated. Retrograde is to the right. Scale bar, 2 μm. Playback at five frames per second. Movie stills are shown in *Figure 3F*.

DOI: https://doi.org/10.7554/eLife.44219.008

in neurons from both young adult and aged mice (*Figure 3—figure supplement 1*, *Videos 5* and *6*). These data suggest that aging does not impair the initial steps of autophagosome biogenesis. However, there is a striking block in LC3B recruitment downstream from the recruitment of both ATG13 and ATG5 that occurs infrequently in neurons from young adult mice, but predominates in neurons from aged mice.

## Stalled AVs recruit autophagosome biogenesis components

To further characterize stalled events in neurons from aged mice, we asked if other autophagy components colocalize with stalled AVs. ATG9 is the only multi-pass transmembrane protein in the core autophagy machinery (*Lang et al., 2000*; *Noda et al., 2000*; *Young et al., 2006*) and is thought to transit to the growing isolation membrane with donor membranes (*Sekito et al., 2009*; *Suzuki et al., 2015*; *Yamamoto et al., 2012*; *Young et al., 2006*). Normally, ATG9 is only transiently associated with the developing autophagosome (*Koyama-Honda et al., 2013*; *Orsi et al., 2012*). We used multi-color live-cell imaging to assess colocalization between autophagy components in neurons from young adult or aged mice co-expressing fluorescently labeled LC3B, ATG9, and ATG13 or ATG5 (*Figure 4A*). As expected, we did not observe significant colocalization of SNAP-

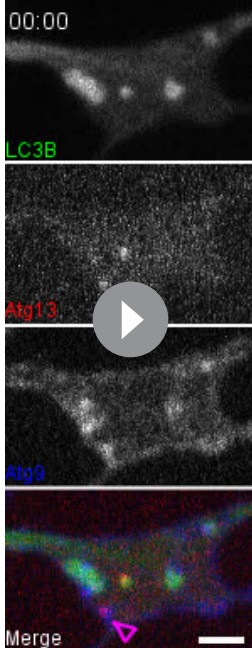

**Video 5.** Productive and stalled biogenesis events occur in the same neuron from a young adult mouse. GFP-LC3B, mCh-ATG13, and SNAP-ATG9 in the distal neurite of a DRG neuron from a young adult mouse depicting both productive (solid arrowhead, solid arrow) and stalled (outlined arrowhead) autophagosome biogenesis events. In the merge movie (bottom), arrowhead color denotes colocalization state of indicated punctum. Retrograde is to the right. Scale bar, 2 µm. Playback at five frames per second. Movie stills are shown in *Figure 3—figure supplement 1A*.

DOI: https://doi.org/10.7554/eLife.44219.010

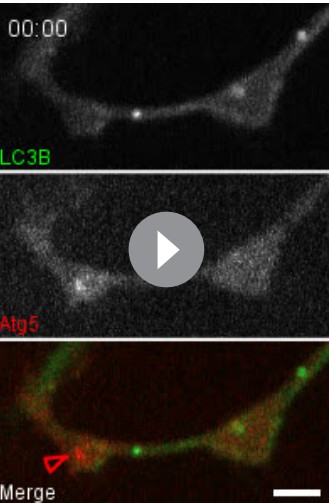

**Video 6.** Productive and stalled biogenesis events occur in the same neuron from an aged mouse. GFP-LC3B and mCh-ATG5 in the distal neurite of a DRG neuron from an aged mouse depicting both productive (solid arrowhead) and stalled (outlined arrowhead) autophagosome biogenesis events. In the merge movie (bottom), arrowhead color denotes colocalization state of indicated punctum. Retrograde is to the right. Scale bar, 2 µm. Playback at five frames per second. Movie stills are shown in *Figure 3—figure supplement 1B*.

DOI: https://doi.org/10.7554/eLife.44219.011

ATG9 with productive autophagosome biogenesis events in neurons from either young or aged mice (*Figure 4B and D*). However, we did observe the robust and persistent colocalization of SNAP-ATG9 with mCh-ATG13 or mCh-ATG5 in the majority of stalled events in neurons from aged mice (*Figure 4A,C and D*, *Video 4,7*). Further, we noted persistent SNAP-ATG9 colocalization with the very rare stalled events seen in neurons from young adult mice (*Figure 4D*). These data suggest that ATG9 may only transiently associate with productive biogenesis events, whereas the majority of stalled AVs aberrantly accumulate or retain ATG9.

We next sought evidence for a similar stalling of autophagosome biogenesis in vivo. As above, we used NMJs to examine AVs in intact tissues. We identified LC3-positive AVs at these synapses in muscle tissue dissected from both young adult and aged mice (*Figure 4E*). We used ATG9 colocalization with either ATG13 or ATG5 as a marker for stalled events in fixed tissue. In NMJs from young adult mice, we observed ATG13 puncta within the presynaptic compartment, but those puncta did not colocalize with ATG9 (*Figure 4F*, left). In contrast, in NMJs from aged mice, we observed the colocalization of ATG13 with ATG9, indicating the persistence of stalled AV formation within the presynaptic compartment in vivo (*Figure 4F*, right). We then quantified the number of stalled AVs in the NMJ motor axon terminal. While we observed stalled AVs (using ATG9 colocalization with ATG13 as a stalled AV marker) only rarely in motor axon terminals from young adult mice, NMJ axon terminals from aged mice consistently contained several stalled AVs (*Figure 4G*). In contrast, we did not detect a change with age in the number of ATG13-positive puncta that did not co-recruit ATG9 (*Figure 4H*). Thus, the fraction of stalled AVs to total ATG13-positive puncta significantly increased with age (*Figure 4I*). These data suggest that our observations of stalled events in cultured primary DRG neurons from aged mice can also be seen in other neuronal types in vivo.

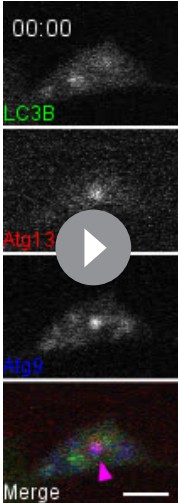

**Video 7.** Atg9 accumulates at stalled biogenesis events in a neuron from an aged mouse. GFP-LC3B, mCh-ATG13, and SNAP-ATG9 in the distal neurite of a DRG neuron from an aged mouse depicting a stalled autophagosome biogenesis event. In the merge movie (bottom), arrowhead color denotes colocalization state of the punctum. Retrograde is to the right. Scale bar, 2 μm. Playback at five frames per second. Movie stills are shown in *Figure 4A*.
DOI: https://doi.org/10.7554/eLife.44219.014

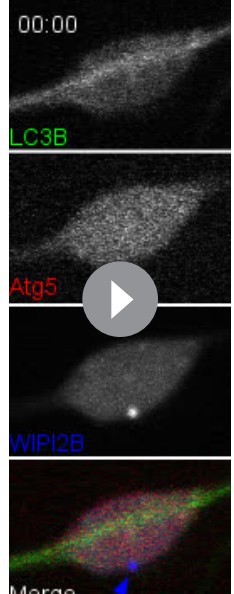

**Video 8.** WIPI2B colocalizes with other autophagy components in a productive event in a neuron from an aged mouse. GFP-LC3B, mCh-ATG13, and Halo-WIPI2B in the distal neurite of DRGs from aged mice depicting a productive autophagosome biogenesis event. In the merge movie (bottom), arrowhead color denotes colocalization state of the punctum. Retrograde is to the right. Scale bar, 2 μm. Playback at five frames per second. Movie stills are shown in *Figure 6E*.
DOI: https://doi.org/10.7554/eLife.44219.022

Since LC3B is not recruited to stalled AVs, we asked whether this defect was due to a failure to recruit the elongation stage constituents required for LC3B lipidation. Using multi-color immunocytochemistry, we examined the localization of endogenous elongation stage components ATG12, ATG7, ATG16L1, and ATG3. We used colocalization of endogenous ATG9 with ATG5 to identify stalled AVs in fixed neurons from aged mice. We observed that the lipidation machinery was successfully recruited to stalled AVs (*Figure 5A–E*).

Given the lack of LC3B recruitment to stalled AVs harboring intact lipidation machinery, we asked whether other LC3B homologs could be recruited to stalled AVs in aged mice. There are multiple orthologs of yeast Atg8 expressed in mammals (mAtg8s), including LC3A, LC3B, LC3C, γ-aminobutyric acid receptor-associated protein (GABARAP), GABARAP-Like 1 (GABARAPL1/GEC1), and GABARAPL2/GATE16 (*Schaaf et al., 2016*). Mice do not appear to have a gene encoding LC3C, but may express a LC3 isoform related to human LC3C (*Liu et al., 2017*). Both immunocytochemistry (*Figure 5—figure supplement 1A–D*) and live cell imaging (*Figure 5F–I*) revealed that LC3A, GABARAP, GABARAPL1/GEC1, and GABARAPL2/GATE16 can each associate with stalled AVs in neurons from aged mice, with each mCherry-mAtg8 colocalizing with persistent Halo-ATG5 puncta (*Figure 5J*). These data indicate that the deficit in LC3B recruitment to stalled AVs in neurons from aged mice is specific and that the recruitment of other mAtg8s is not sufficient to convert stalled AVs into productive AVs in neurons from aged mice. These observations are consistent with a growing literature indicating that mAtg8s are not fully functionally redundant (*Nguyen et al., 2016*).

Also using live-cell imaging, we asked if ectopic expression of the mAtg8s altered the assembly kinetics of AVs in neurons from aged mice. Surprisingly, we found that overexpression of mScarlet-LC3A, but not other mAtg8s, caused GFP-LC3B recruitment to 84.2% of stalled AVs (persistent Halo-ATG5 puncta), significantly different from control neurons (*Figure 5—figure supplement 1E*; p=0.0002; 21.1% of stalled AVs in control). However, this induced recruitment of GFP-LC3B to stalled AVs did not resolve the stalled event (data not shown), further implying that while the failure to recruit LC3B is a hallmark of stalled AV events, it is not the principal defect involved.

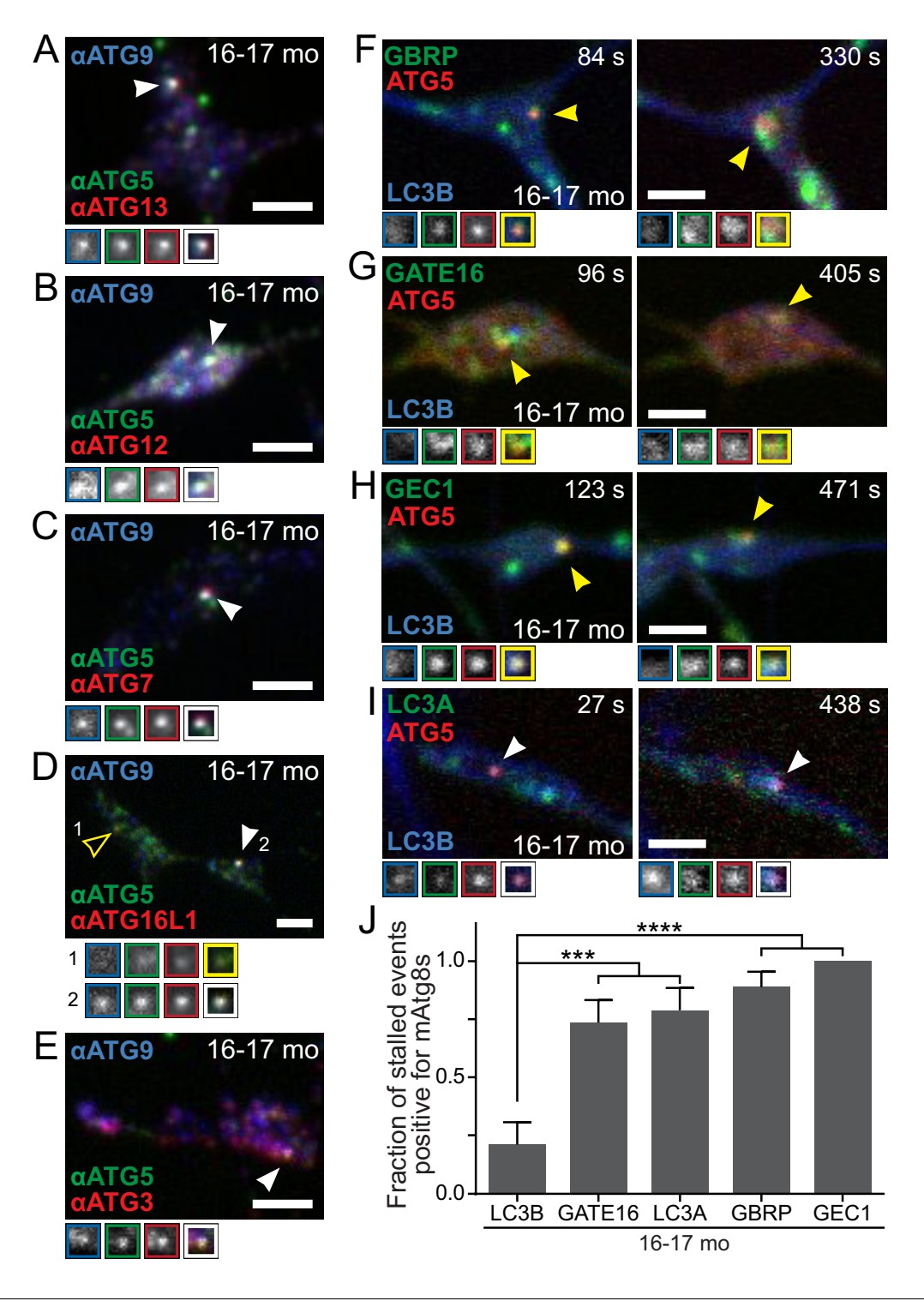

**Figure 5.** Recruitment of autophagy machinery, including LC3B homologs, to stalled AVs is not sufficient to rescue the biogenesis defect. (A–E) Representative maximal projection micrographs of the distal neurites of fixed DRG neurons from aged mice. Stalled AVs are identified by colocalization of anti-ATG5 (green) and anti-ATG9 (blue). Antibodies to other autophagy components are visualized in red: ATG13 (A) or elongation complex components ATG12 (B), ATG7 (C), ATG16L1 (D), and ATG3 (E). Arrowheads denote colocalization state of AVs (stalled, filled arrowheads). Borders of magnifications of indicated puncta denote channel or colocalization state in merge. (F–I) Time series (channels merged) of Halo-ATG5, mAtg8s (mCh-GABARAP in F, mCh-GABARAPL2/GATE16 in G,

*Figure 5 continued on next page*

*Figure 5 continued*
mCh-GABARAPL1/GEC1 in H, and mScarlet-LC3A in I), and GFP-LC3B in the distal neurite of DRG neurons from aged mice depicting stalled AVs. Arrowheads denote colocalization of mAtg8s with stalled AVs. Magnified views of denoted puncta are shown below full micrograph; border color represents channel or colocalization state in merge. Time is indicated as time since stalled AV was first visible. (J) Quantification of the fraction of stalled AVs co-recruiting each LC3/GABARAP family member when individually overexpressed in DRGs from aged mice (mean ± SEM; n ≥ 10 stalled AVs in three biological replicates for each mAtg8). ***p<0.001; ****p<0.0001 by two-tailed Fisher's exact test. Scale bars, 2 µm.
DOI: https://doi.org/10.7554/eLife.44219.017
The following figure supplement is available for figure 5:

**Figure supplement 1.** Endogenous mAtg8s are recruited to stalled AVs in vitro.
DOI: https://doi.org/10.7554/eLife.44219.018

## Overexpression of WIPI2 restores the rate of autophagosome biogenesis in neurons from aged mice

PROPPINs (β-propellers that bind phosphoinositides) are essential PI3P effectors in autophagy and are conserved from yeast to humans (*Michell et al., 2006*; *Polson et al., 2010*; *Proikas-Cezanne et al., 2004*). In mammals, there are four PROPPINs, termed WD-repeat protein interacting with phosphoinositides (WIPI1 through WIPI4) (*Polson et al., 2010*; *Proikas-Cezanne et al., 2004*). WIPI1 and WIPI2 are closely related and orthologs of yeast Atg18, while WIPI3 and WIPI4 form a separate paralogous group (*Behrends et al., 2010*; *Polson et al., 2010*; *Proikas-Cezanne et al., 2004*). WIPI1, the first family member to be identified to have a role in autophagy, is recruited to autophagosomal membranes upon autophagy induction (*Gaugel et al., 2012*; *Itakura and Mizushima, 2010*; *Proikas-Cezanne et al., 2007*; *Proikas-Cezanne et al., 2004*; *Vergne et al., 2009*). WIPI2 links PI3P production by the autophagy nucleation complex to LC3 interaction with the isolation membrane as WIPI2 binds to both PI3P and ATG16L1 (*Figure 6A*) (*Dooley et al., 2014*; *Lamb et al., 2013*; *Polson et al., 2010*). Thus, we hypothesized that alterations in WIPI2 function may result in lower levels of LC3B recruitment and deleteriously affect productive biogenesis.

First we confirmed the importance of WIPI2 in autophagosome biogenesis in primary neurons. Depletion of WIPI2 by RNAi (*Figure 6B*) did not alter rates of AV initiation (determined by mCh-ATG5 puncta generation) (*Figure 6C*), but led to a significant deficit in autophagosome biogenesis, which was fully restored by expression of an RNAi-resistant human Halo-WIPI2B construct (*Figure 6D*). WIPI2 binds PI3P via a conserved FRRG motif (*Baskaran et al., 2012*; *Dove et al., 2004*; *Gaugel et al., 2012*; *Jeffries et al., 2004*; *Krick et al., 2006*; *Proikas-Cezanne et al., 2007*; *Proikas-Cezanne et al., 2004*; *Watanabe et al., 2012*). This interaction can be abolished by mutating the positively charged arginine residues in the motif to uncharged threonine residues (FTTG) (*Figure 6A*) (*Dooley et al., 2014*). Overexpression of Halo-WIPI2B(FTTG) was unable to rescue the deficit, consistent with a key role for phosphoinositide signaling in autophagosome biogenesis (*Figure 6D*). WIPI2B also interacts with ATG16L1, an essential component of the LC3 conjugation complex. The interaction between WIPI2B and ATG16L1 can be abrogated by switching a positively charged arginine to a negatively charged glutamate (R108E) in WIPI2B (*Dooley et al., 2014*). Ectopic expression of Halo-WIPI2B(R108E) in WIPI2-depleted neurons from young adult mice did not affect rates of AV initiation but did not rescue the deficit in the rates of formation of GFP-LC3B-positive autophagosomes (*Figure 6C–D*). Furthermore, overexpression of the WIPI2 paralog SNAP-WIPI1A was also unable to compensate for the loss of WIPI2 in young adult neurons (*Figure 6D*). These data confirm that WIPI2, including its known functional domains, is required in autophagosome biogenesis in DRG neurons.

Next, we ectopically expressed Halo-tagged WIPI2B in neurons from aged mice. Halo-WIPI2B colocalized with the early autophagosome marker mCh-ATG5 in neurons (*Figure 6E*, *Video 8*) and did not affect rates of AV initiation (*Figure 6F*). Strikingly, ectopic WIPI2B expression increased rates of autophagosome biogenesis in neurons from aged mice from 0.21 AVs per minute to 0.47 AVs per minute (*Figure 6G*), a rate similar to that observed in neurons from young adult mice (*Figure 1B*). Furthermore, overexpression of WIPI2B did not alter the kinetics of productive AV biogenesis (*Figure 6H*). In contrast to wild type WIPI2B, expression of WIPI2B constructs with targeted mutations in either the PI3P or ATG6L1 binding motifs did not restore autophagosome biogenesis in

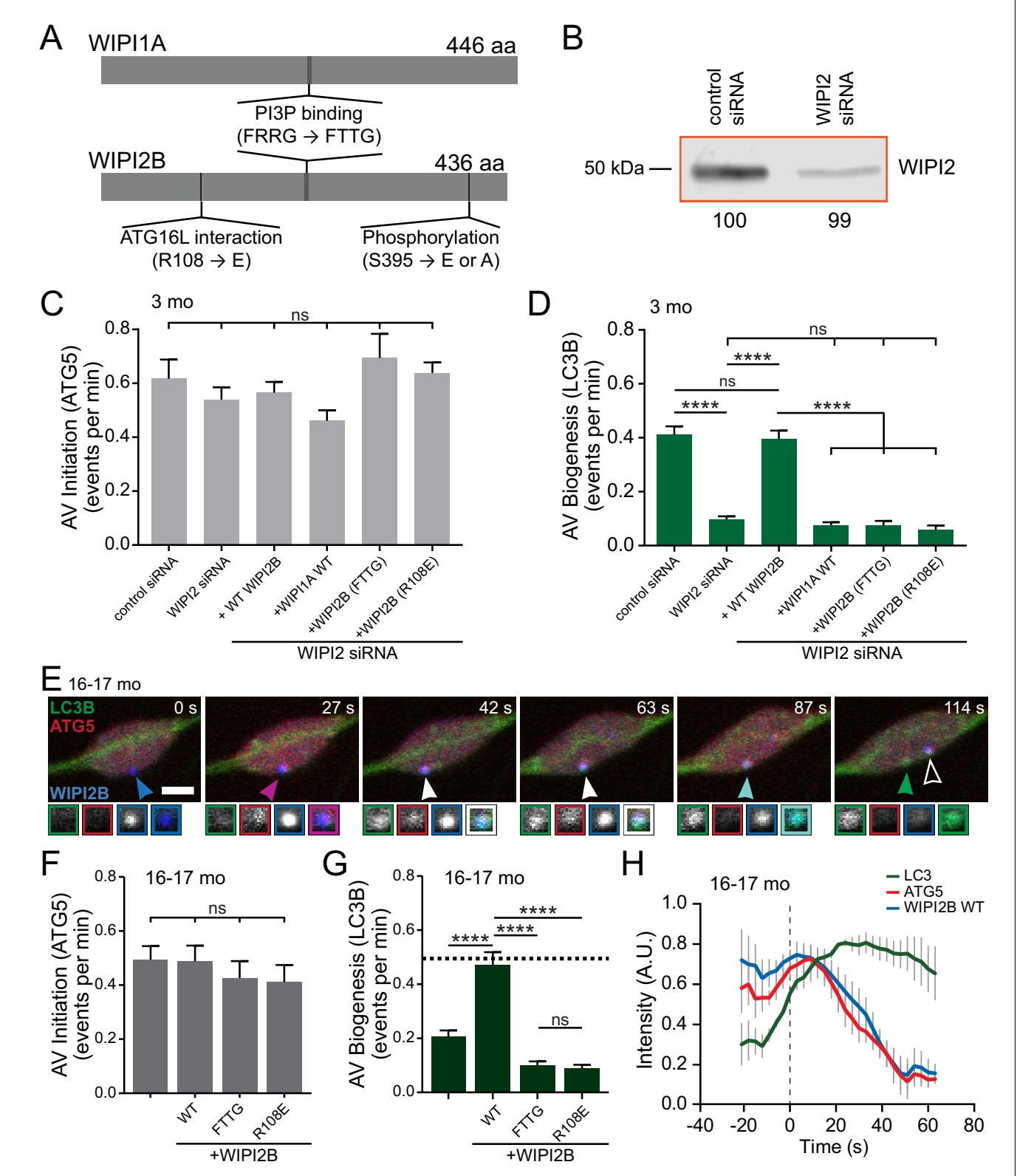

**Figure 6.** Overexpression of WIPI2B in neurons from aged mice returns autophagosome biogenesis to levels observed in neurons from young adult mice. (**A**) Schematic of WIPI1A and WIPI2B proteins, depicting the PI3P-interaction domains (FRRG), the ATG16L1-binding domain in WIPI2B (R108), and the phosphorylation site in WIPI2B (S395). Point mutations to disrupt these interactions are indicated in the relevant domains (FRRG → FTTG, R108 → E, or S395 → E or A). (**B**) Immunoblot of DRG lysates treated with indicated siRNA, collected after 2 days in vitro. Total protein was used as a loading

*Figure 6 continued on next page*

Figure 6 continued

control; normalization factor is indicated below blot as a percentage. (C) Quantification of the rate of AV initiation (mCh-ATG5 puncta) in DRG neurons from young adult mice with control or WIPI2 siRNA or WIPI2 siRNA with indicated RNAi-resistant Halo-WIPI2B or SNAP-WIPI1A constructs (mean ± SEM; n ≥ 15 neurons from three biological replicates for each siRNA condition). ns (not significant) p>0.05 by Kruskal-Wallis ANOVA test with Dunn's multiple comparisons test. (D) Quantification of the rate of AV biogenesis (GFP-LC3B puncta) in DRG neurons from young adult mice with control or WIPI2 siRNA or WIPI2 siRNA with indicated RNAi-resistant Halo-WIPI2B or SNAP-WIPI1A constructs (mean ± SEM; n ≥ 15 neurons from three biological replicates for each siRNA condition). ns p>0.05; ****p<0.0001 by Kruskal-Wallis ANOVA test with Dunn's multiple comparisons test. (E) Time series of merged micrographs of GFP-LC3B, mCh-ATG5, and Halo-WIPI2B WT in the distal neurite of a DRG neuron from an aged mouse depicting a productive autophagosome biogenesis event. Arrowheads indicate colocalization state on the isolation membrane; solid arrowhead follows one punctum, while outlined arrowhead indicates a different AV biogenesis event. Magnified views of denoted puncta are shown below full micrograph; border color represents channel or colocalization state in merge. Retrograde is to the right. Scale bar, 2 μm. (F) Quantification of the rate of AV initiation (mCh-ATG5 puncta) in DRG neurons from aged mice with or without overexpression of the indicated Halo-WIPI2B construct (mean ± SEM; n ≥ 17 neurons from three biological replicates for each condition). ns p>0.05 by Kruskal-Wallis ANOVA test with Dunn's multiple comparisons test. (G) Quantification of the rate of AV biogenesis (marked with GFP-LC3B) in DRG neurons from aged mice with or without overexpression of the indicated Halo-WIPI2B construct (mean ± SEM; n ≥ 17 neurons from three biological replicates for each condition). ****p<0.0001; ns p>0.05 by Kruskal-Wallis ANOVA test with Dunn's multiple comparisons test. (H) Mean intensity profiles of mCh-ATG5 (red), Halo-WIPI2B WT (blue), and GFP-LC3B (green) for productive AVs (mean ± SEM; n = 6 biogenesis events from five neurons from three biological replicates). Vertical dashed line indicates the half-maximum of GFP-LC3B intensity, which was used to align the traces.
DOI: https://doi.org/10.7554/eLife.44219.019

The following figure supplements are available for figure 6:

**Figure supplement 1.** WIPI protein and mRNA levels.
DOI: https://doi.org/10.7554/eLife.44219.020
**Figure supplement 2.** Autophagy protein levels.
DOI: https://doi.org/10.7554/eLife.44219.021

neurons from aged mice (*Figure 6F–G*). These data suggest that overexpression of WIPI2B in neurons from aged mice restores autophagosome biogenesis and that this rescue requires both the PI3P-binding and ATG16L1-binding functions of WIPI2B.

## WIPI2B phosphorylation is a molecular switch regulating autophagosome biogenesis

Given that ectopically expressing WIPI2B in neurons from aged mice rescues rates of autophagosome biogenesis, we initially hypothesized that WIPI2 levels decrease in neuronal tissues with age. However, we observed no significant deficits in WIPI2 expression levels at the level of RNA or protein, or those of any of the WIPI family members with age in either whole brain lysates or DRG lysates (*Figure 6—figure supplement 1*). We also observed no significant changes in expression levels of several other autophagy components (ULK1, P-ULK1, ATG14, P-ATG14, Beclin1, ATG3, ATG5, ATG7, ATG10, ATG16L1, LC3B, LC3A, GABARAP, GABARAPL1, GABARAPL2, p62, WIPI3, and WIPI4) with age in whole brain or DRG lysates (*Figure 6—figure supplement 1A and B*, *Figure 6—figure supplement 2*). Additionally, using immunocytochemistry on fixed DRG neurons from aged mice, we observed endogenous WIPI1 and WIPI2 localized to stalled AVs (*Figure 5—figure supplement 1F–G*). These results indicate that the decrease in autophagosome biogenesis with age is not due to an age-related loss of WIPI2 or its paralogs and that endogenous WIPIs can be recruited to stalled AVs in neurons from aged mice.

Next we looked to post-translational modification of WIPI2. WIPI2B is known to be phosphorylated at serine 395 (S413 in WIPI2A) (*Hsu et al., 2011*; *Wan et al., 2018*), although the mechanistic effects of this phosphorylation have not been fully explored. We used two independent phosphorylation-sensitive antibodies (*Figure 7—figure supplement 1A,C and D*) to confirm that phosphorylated WIPI2 is found in neuronal tissues (*Figure 7A*). Additionally, we confirmed that we could detect phospho-WIPI2 on AVs in DRG distal neurites by immunocytochemistry (*Figure 7B*, *Figure 5—figure supplement 1H*). Our data (*Figure 7—figure supplement 1B*) agreed with a previous report (*Wan et al., 2018*) that phosphorylation of WIPI2B at serine 395 does not affect its ability to bind to PI3P or Atg16L1. Next, we asked how WIPI2 phosphorylation affects autophagosome biogenesis. We ectopically expressed a RNAi-resistant nonphosphorylatable construct, Halo-WIPI2B (S395A), or a RNAi-resistant phospho-mimetic construct, Halo-WIPI2B(S395E), in WIPI2-depleted neurons from young adult mice. Similar to our previous results, overexpression of Halo-WIPI2B

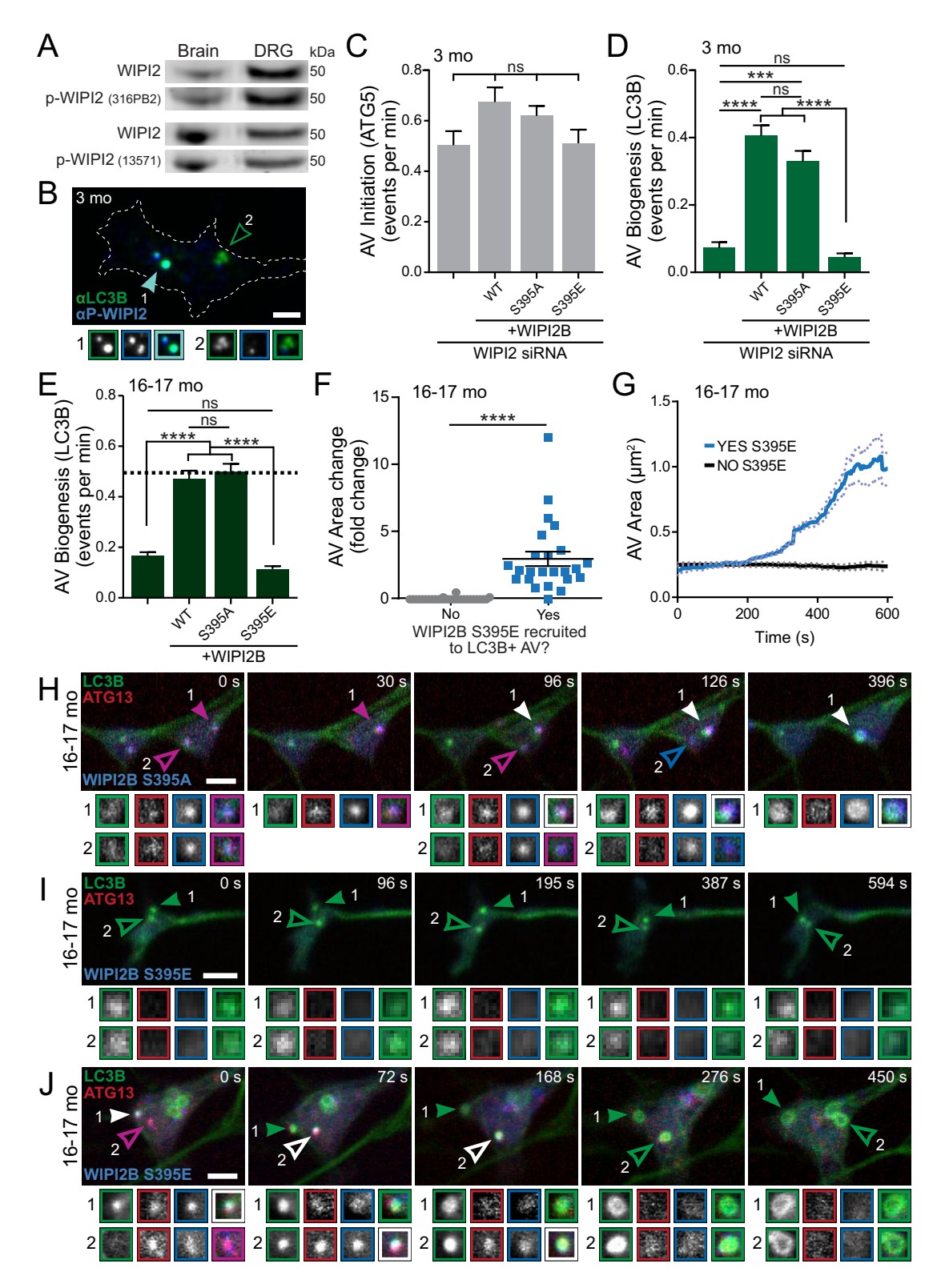

**Figure 7.** Dynamic WIPI2B phosphorylation in neurons is required for autophagosome biogenesis. (**A**) Immunoblot of whole brain or DRG lysates from aged mice demonstrates p-WIPI2 can be detected with two different phospho-WIPI2 antibodies. (**B**) Representative maximal projection immunocytochemistry micrograph of LC3B and phospho-WIPI2 in the distal tip of a DRG neuron from a young adult mouse. Arrowheads indicate colocalization state on the AV; solid arrowhead indicates puncta with both proteins, while outlined arrowhead designates a different AV with no

*Figure 7 continued*

phospho-WIPI2 colocalized with LC3B. Borders of magnifications of indicated puncta denote channel or colocalization state in merge. Scale bar, 2 μm. (C) Quantification of the rate of AV initiation (mCh-ATG5 puncta) in DRG neurons from young adult mice with WIPI2 siRNA or WIPI2 siRNA with indicated RNAi-resistant Halo-WIPI2B constructs (mean ± SEM; n ≥ 17 neurons from three biological replicates for each siRNA condition). ns (not significant) p>0.05 by Kruskal-Wallis ANOVA test with Dunn's multiple comparisons test. (D) Quantification of the rate of AV biogenesis (GFP-LC3B puncta) in DRG neurons from young adult mice with WIPI2 siRNA or WIPI2 siRNA with indicated RNAi-resistant Halo-WIPI2B constructs (mean ± SEM; n ≥ 17 neurons from three biological replicates for each siRNA condition). ns (not significant) p>0.05; ***p=0.0001; ****p<0.0001 by Kruskal-Wallis ANOVA test with Dunn's multiple comparisons test. (E) Quantification of the rate of AV biogenesis (marked with GFP-LC3B) in DRG neurons from aged mice with or without overexpression of the indicated Halo-WIPI2B construct (mean ± SEM; n ≥ 23 neurons from three biological replicates for each condition). ****p<0.0001; ns (not significant) p>0.05 by Kruskal-Wallis ANOVA test with Dunn's multiple comparisons test. Horizontal dashed line indicates rate of AV biogenesis in neurons from young adult mice. (F) Quantification of AV area change with or without recruitment of Halo-WIPI2B (S395E) (mean ± SEM; n ≥ 24 AVs from 14 neurons from three biological replicates). ****p<0.0001 by Mann-Whitney test. (G) Individual AV area profiles were averaged to improve signal-to-noise of GFP-LC3B puncta that were positive (blue) or negative (black) for Halo-WIPI2B(S395E) (mean ± SEM; n ≥ 25 AVs from ≥17 neurons from three biological replicates). (H) Time series of merge micrographs of GFP-LC3B, mCh-ATG13, and Halo-WIPI2B (S395A) in the distal neurite of a DRG neuron from an aged mouse. (I–J) Time series of merged micrographs of GFP-LC3B, mCh-ATG13, and Halo-WIPI2B(S395E) in the distal neurite of DRG neurons from aged mice depicting AVs that fail to recruit Halo-WIPI2B(S395E) and fail to grow (I) and AVs that do recruit Halo-WIPI2B(S395E) and increase in area (J) during the imaging window. Arrowheads indicate colocalization state on the isolation membrane; solid arrowhead follows one punctum, while outlined arrowhead indicates a second AV. Magnified views of denoted puncta are shown below full micrograph; border color represents channel or colocalization state in merge. Retrograde is to the right. Scale bars, 2 μm.

DOI: https://doi.org/10.7554/eLife.44219.023

The following figure supplements are available for figure 7:

**Figure supplement 1.** WIPI2B is phosphorylated at S395.
DOI: https://doi.org/10.7554/eLife.44219.024

**Figure supplement 2.** GFP-LC3B-positive AVs do not enlarge when Halo-WIPI2B(S395E) is not recruited.
DOI: https://doi.org/10.7554/eLife.44219.025

constructs did not affect rates of AV initiation (*Figure 7C*). We found that the phospho-dead construct, Halo-WIPI2B(S395A), rescued rates of autophagosome biogenesis similar to the wild type Halo-WIPI2B construct. In contrast, overexpression of the phospho-mimetic construct, Halo-WIPI2B (S395E), did not restore rates of autophagosome biogenesis in WIPI2-depleted neurons from young adult mice (*Figure 7D*). When we expressed these constructs in neurons from aged mice, we obtained similar results; the phospho-dead construct, Halo-WIPI2B(S395A), restored the rate of autophagosome biogenesis in neurons from aged mice, while the phospho-mimetic construct, Halo-WIPI2B(S395E), did not (*Figure 7E*). We did not detect differences in the levels of overexpression for the different Halo-WIPI2B constructs or changes in overexpression of ectopic Halo-WIPI2B constructs with age (*Figure 7—figure supplement 1E*). These data suggest that WIPI2 must be dephosphorylated to enable productive AV biogenesis.

These results led us to hypothesize that levels of phosphorylated WIPI2 increase with age in neuronal tissues. However, just as we saw with total WIPI2 levels, we did not see an overall change in phosphorylated WIPI2 protein with age in either whole brain or DRG lysates (*Figure 6—figure supplement 1C–E*). These results suggest that WIPI2 may be found in its phosphorylated form throughout the cytosol and only transiently dephosphorylated at the isolation membrane, masking any functional age-related change in WIPI2 phosphorylation in bulk assays. This hypothesis is consistent with our data indicating that stalled and productive AVs occur in the same axonal tip (*Figure 3—figure supplement 1*, *Video 6*), suggesting that AV stalling results from a highly localized defect.

To further characterize the role of phosphorylated WIPI2B in autophagosome biogenesis, we examined AV events in neurons from aged mice overexpressing the phospho-dead construct, Halo-WIPI2b(S395A) or the phospho-mimetic construct, Halo-WIPI2B(S395E) in conjunction with GFP-LC3B and mCh-ATG13, by live-cell microscopy. The phospho-dead construct Halo-WIPI2B(S395A) was cytoplasmic, but also colocalized with mCh-ATG13 and GFP-LC3B on productive AVs (*Figure 7H*). Next, we examined neurons from aged mice ectopically expressing the phospho-mimetic Halo-WIPI2B(S395E) construct. We identified GFP-LC3B-positive AVs and determined whether WIPI2B(S395E) was ever associated with the AV during the video. GFP-LC3B-positive AV events that failed to recruit Halo-WIPI2B(S395E) did not increase in size (*Figure 7F,G and I*, *Figure 7—figure supplement 2*), suggesting that the isolation membrane could not successfully extend. In contrast, GFP-LC3B-positive AV events that did recruit Halo-WIPI2B(S395E) increased in

size to form a GFP-LC3B ring structure, consistent with a mature autophagosome (*Figure 7F,G and J*). Taken together, these results suggest that WIPI2B is dephosphorylated at the isolation membrane to allow autophagosome biogenesis to initiate. Further, these results suggest that WIPI2B is then dynamically rephosphorylated at the AV to enable the autophagosome to grow and complete biogenesis.

If this hypothesis is correct, phosphorylation of WIPI2B at serine 395 might affect its affinity for membranes. To determine whether the phosphorylation state of WIPI2B affects its ability to interact with membranes, we performed crude fractionation experiments. We collected whole brain lysates from young adult and aged nontransgenic mice and separated the cytosolic and membrane fractions by centrifugation. We then compared the endogenous levels of phospho-WIPI2 and total WIPI2 associated with each fraction by immunoblot (*Figure 8A*). The ratio between phospho-WIPI2 and total WIPI2 in the membrane fraction was reduced by approximately 50% compared to the cytosolic fraction for both ages (*Figure 8B*). These results suggest that phosphorylation of WIPI2B at serine 395 decreases its affinity for membranes.

We also tested our hypothesis that phosphorylation of WIPI2B at serine 395 causes WIPI2B to disassociate from the AV membrane in live-cell imaging of both the phospho-dead WIPI2B(S395A) and phospho-mimetic WIPI2B(S395E) constructs in the same neurons. We examined the dynamics of ectopically expressed RNAi-resistant Halo-WIPI2B(S395A) and SNAP-WIPI2B(S395E) in WIPI2-depleted DRG neurons from young adult mice. We measured the length of time each WIPI2B construct resided at a given punctum. Since our time-lapse videos were captured over 10 min, the maximum residence time we could measure was 600 s. We found that Halo-WIPI2B(S395A) remained associated for nearly 600 s on average. In contrast, in the same neurons, SNAP-WIPI2B(S395E) only remained associated with puncta for approximately 300 s, or half as long as the phospho-dead construct (*Figure 8C*). These data are consistent with our lysate fractionation data, indicating that phosphorylation of WIPI2B at serine 395 correlates with a decreased affinity for membranes.

Similar to our observations of other aspects of AV biogenesis, within a given DRG axonal tip, we could observe both an expanding AV that co-recruited Halo-WIPI2B(S395E) and an AV that was Halo-WIPI2B(S395E)-negative that failed to enlarge (*Figure 8D*). Thus, the rephosphorylation of WIPI2B(S395) may also be a highly localized process. The dynamic phosphorylation of WIPI2 during AV biogenesis at the isolation membrane could allow for tight, spatially and temporally localized regulation of autophagy (*Figure 9*).

## Discussion

Here, we investigated the dynamics of autophagy during aging in primary neurons and demonstrated that the rate of autophagosome biogenesis significantly decreases in neurons with age. Surprisingly, the deficit was specific, as the initial stages of autophagosome formation, initiation and nucleation, were not altered with age in mammalian neurons. Instead, we found that the majority of AVs in aged neurons successfully initiated but then stalled. EM analysis suggested that this deficit was correlated with a morphological defect in autophagosome formation, characterized by excess membrane accumulation within the autophagic vacuole, detectable in aged neurons both in vitro and in vivo. WIPI2B overexpression in neurons from aged mice increased the rate of autophagosome biogenesis, restoring this rate to that found in neurons from young adult mice. Further, we propose that the dynamic regulation of WIPI2B phosphorylation at the isolation membrane may be integral to autophagosome biogenesis. Our results indicated that the nonphosphorylatable S395A form of WIPI2 was sufficient to rescue AV biogenesis upon depletion of endogenous WIPI2, while recruitment of the phosphomimetic WIPI2B(S395E) mutant correlated with expansion of the nascent autophagosome, suggesting that both dephosphorylation and rephosphorylation of WIPI2B are key regulatory steps. Ultimately, we showed that the rate of autophagosome biogenesis decreased in neurons during aging, but we mitigated this decrease by overexpressing a single autophagy component, WIPI2B (*Figure 6G*).

In neurons from aged mice, the majority of stalled AVs aberrantly accumulated ATG9 (*Figure 4*). Our results showing that WIPI2B overexpression in aged neurons restored autophagosome biogenesis are consistent with previous studies indicating that WIPI2 downregulation induced the localized accumulation of ATG9 at AVs (*Orsi et al., 2012*). We hypothesize that the multilamellar structures we detected by TEM (*Figure 1*) correlate with the stalled AVs we observed by fluorescence

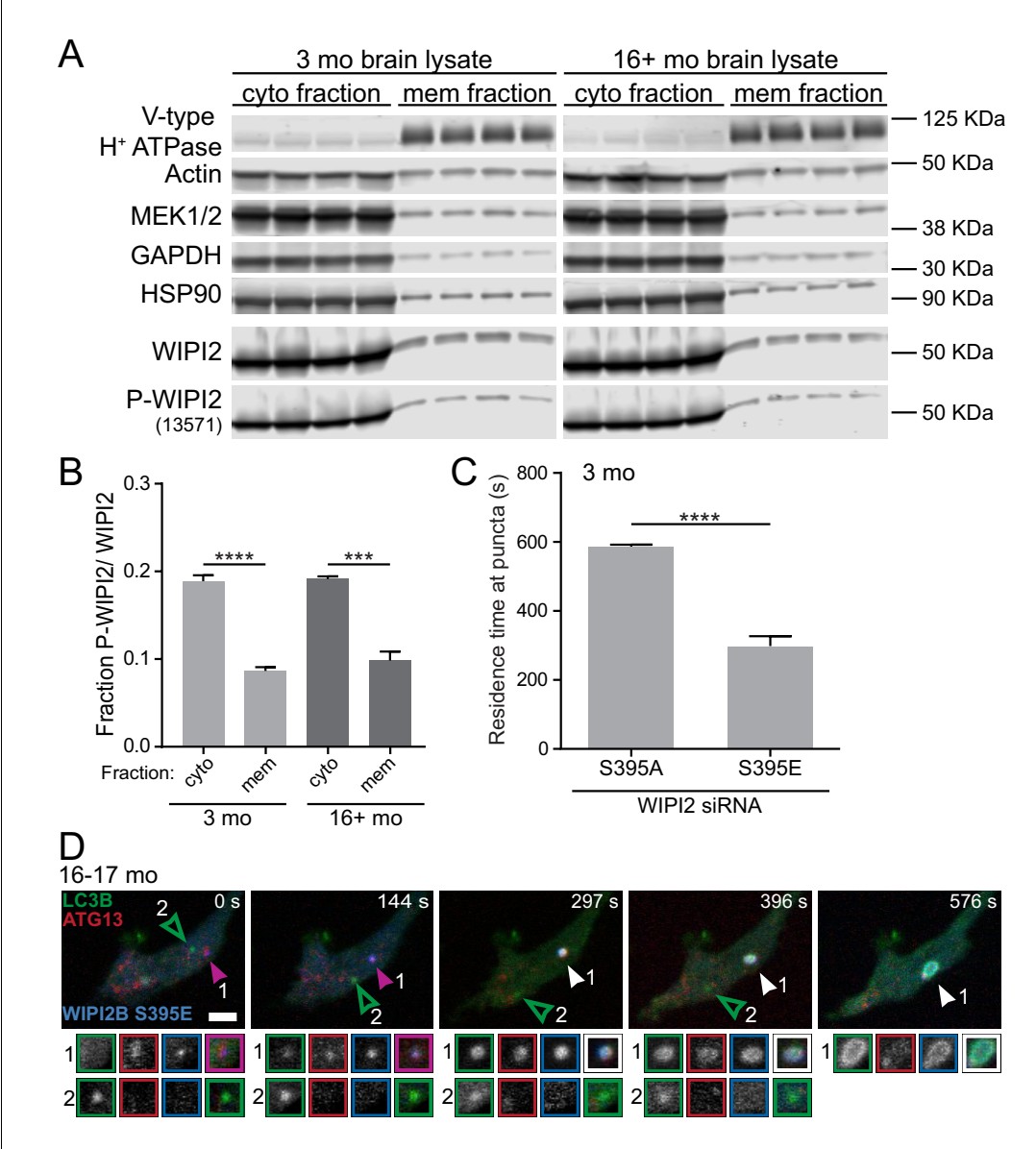

**Figure 8.** Phosphorylation of WIPI2B at serine 395 decreases the affinity of WIPI2B for membranes. (**A**) Immunoblot of cytosolic and membrane fractions of brain lysates from young adult (left) and aged (right) mice (n = 4 biological replicates for each age). (**B**) Quantification of the ratio of Phospho-WIPI2 to total WIPI2 in the cytosolic (cyto) and membrane (mem) fractions from the immunoblot in A (mean ± SEM; n = 4 biological replicates for each age). ****p<0.0001; ***p=0.0001 by two-tailed unpaired t test. (**C**) Quantification of the residence time of Halo-WIPI2B(S395A) or SNAP-WIPI2B(S395E) in the distal tips of DRG neurons from young adult mice with WIPI2 siRNA (mean ± SEM; n = 25 neurons from three biological replicates). ****p<0.0001 by Mann-Whitney test. (**D**) Time series of merged micrographs of GFP-LC3B, mCh-ATG5, and Halo-WIPI2B(S395E) in the distal neurite of a DRG neuron from an aged mouse depicting an AV that fails to recruit Halo-WIPI2B(S395E) and fails to grow (open arrowhead) and an AV that does recruit Halo-WIPI2B(S395E) and increases in area (filled arrowhead) during the imaging window in the same DRG distal neurite. Arrowheads indicate colocalization state on the isolation membrane. Magnified views of denoted puncta are shown below full micrograph; border color represents channel or colocalization state in merge. Retrograde is to the right. Scale bar, 2 μm.

DOI: https://doi.org/10.7554/eLife.44219.026

microscopy. Work from other groups suggest that this hypothesis could be correct. ATG9 interacts with ATG2, a conserved core autophagy protein (*Barth and Thumm, 2001*; *Gómez-Sánchez et al., 2018*; *Shintani et al., 2001*; *Wang et al., 2001*). ATG2 also interacts with WIPI4 (*Behrends et al., 2010*; *Chowdhury et al., 2018*; *Lu et al., 2011*; *Velikkakath et al., 2012*). The speculation that the

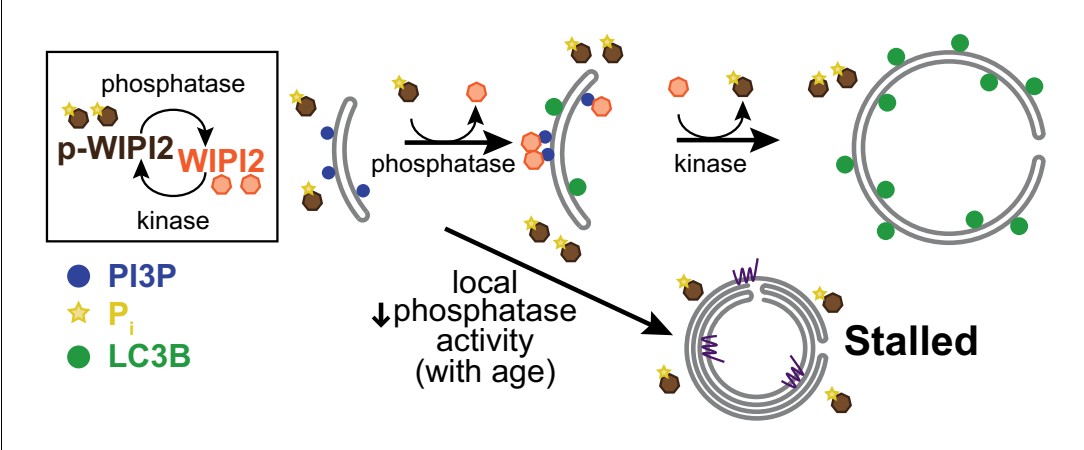

**Figure 9.** Local dynamic phosphorylation of WIPI2B is required for progression through autophagosome biogenesis. Model of how dynamic phosphorylation of WIPI2 and autophagosome biogenesis changes in neurons with aging, characterized by formation of both productive and stalled events.

DOI: https://doi.org/10.7554/eLife.44219.027

ATG2-ATG9 complex transfers lipids to the membrane-hungry growing autophagosome (*Gómez-Sánchez et al., 2018*; *Kumar et al., 2018*) was recently confirmed in yeast (*Osawa et al., 2019*) and mammalian cells (*Valverde et al., 2019*). Thus, the prolonged association of ATG9 with stalled AVs detected in neurons from aged mice (*Figure 4*) may indicate a prolonged association of ATG2 with stalled AVs. This extended residency at the AV could enable unregulated lipid transfer to the stalled AV, resulting in the multilamellar structures we observed in neurons from aged mice (*Figure 1*).

We also found that stalled AVs failed to recruit LC3B (*Figure 3*), while the recruitment of other mAtg8s was not affected (*Figure 5*). Of note, lipidation of LC3B is less efficient on less curved and less PE-rich membranes than lipidation of GABARAPL1 (*Nath et al., 2014*), suggesting that stalled AVs contain sufficient curvature and have a sufficient PE concentration to allow the lipidation and incorporation of all mAtg8s except LC3B. Furthermore, inducing the recruitment of LC3B to AVs by LC3A overexpression was not sufficient to rescue autophagosome formation. These results, in conjunction with recent observations (*Nguyen et al., 2016*; *Tsuboyama et al., 2016*), suggest that LC3B recruitment is neither strictly required nor sufficient for AV generation and elongation. Instead, LC3B recruitment may regulate membrane expansion or membrane fusion to form a double-membrane structure. We propose that upon perturbation of dynamic WIPI2 phosphorylation, membrane extension may proceed in an unrestricted manner, generating the multilamellar structures observed by TEM in neurons from aged mice (*Figure 1*) (*Majeed, 1993*) and in aged human AD brain (*Nixon et al., 2005*). Here, we chose to focus on WIPI2 as a master regulator of autophagosome biogenesis. However, it will be interesting to determine in the future how LC3B incorporation into the isolation membrane relates to lipid incorporation and autophagosome membrane extension and expansion.

The autophagy pathway has been extensively studied in non-neuronal cells, where autophagy can be induced by starvation or other cellular stressors (*Abada and Elazar, 2014*; *Hale et al., 2013*; *Mariño et al., 2011*; *Reggiori and Klionsky, 2013*; *Son et al., 2012*; *Wu et al., 2013*; *Zhang and Baehrecke, 2015*). Conversely, in vivo and in vitro studies in neurons indicate that neuronal autophagy is not significantly induced by starvation (*Fox et al., 2010*; *Maday and Holzbaur, 2016*; *Mizushima et al., 2004*; *Tsvetkov et al., 2010*) or by proteotoxic stress (*Maday et al., 2012*; *Wong and Holzbaur, 2014*). These studies suggest that autophagosome biogenesis is regulated differentially in neuronal and non-neuronal cells. Our results indicate that induction of autophagosome biogenesis is constitutive and remains robust during aging in neurons (*Figure 2*). Further, our data identify a novel age-related regulation of neuronal autophagosome biogenesis, suggesting that autophagy can be regulated at distinct steps apart from the autophagy initiation complex. Consistent with our finding that WIPI2 regulates basal autophagy, WIPI1 and WIPI2 recruitment to AVs is

independent of glucose starvation (*McAlpine et al., 2013*; *Pfisterer et al., 2011*). Our data ultimately suggest that neuronal autophagy may be more easily modulated ectopically via WIPI2B than the better-studied, starvation-sensitive ULK1-ATG13 initiation complex.

Stalled AVs in neurons from aged mice provide a unique opportunity to tease apart the role of WIPI2B in autophagosome biogenesis. WIPI2 interacts with PI3P at the isolation membrane and is required for subsequent WIPI1 localization (*Bakula et al., 2017*). Both WIPI1 and WIPI2 are predicted to form an amphipathic α-helix upon lipid binding, similar to the yeast homolog, Atg18 (*Gopaldass et al., 2017*), which can promote membrane deformation. Further, Atg18 forms oligomers upon membrane binding (*Gopaldass et al., 2017*; *Scacioc et al., 2017*). Considering our data in conjugation with these data from cell lines, we now propose that WIPI2 dephosphorylation at S395 may allow robust recruitment of WIPI2 and WIPI1 to the isolation membrane. The recruitment of WIPI1 and formation of WIPI1-WIPI2 hetero-oligomers promote membrane deformation. Of note, dephosphorylation of yeast Atg18 is required for Atg18 association with the membrane (*Tamura et al., 2013*), suggesting that the dynamic phosphorylation of PROPPINs could be a conserved regulatory mechanism for autophagy.

Further support for a sequential phosphorylation model would come from the identification of the kinase(s) responsible for WIPI2(S395) phosphorylation. One WIPI2 kinase, mTORC1, has recently been identified (*Hsu et al., 2011*; *Wan et al., 2018*). While this study found that phosphorylated WIPI2B led to its degradation by the proteasome in HEK293T cells, our data indicate a more nuanced role for phosphorylated WIPI2 in autophagosome biogenesis in neurons. Our results indicated that phosphorylation at WIPI2B serine 395 lowered the affinity of WIPI2B for membrane and shortened the residence time at the nascent autophagic membrane (*Figure 8A–C*). Recruitment of WIPI2B(S395E) to the isolation membrane correlated with expansion of the autophagosome membrane (*Figure 7F–J*), so we speculate that dissociation of WIPI2B might be a critical and regulated step during autophagosome biogenesis (*Figure 9*). Thus, it will be interesting to determine how a specific post-translational modification could have such opposite effects on the same pathway.

We also propose that a localized phosphatase may regulate WIPI2 activity at the developing autophagosome. One compelling candidate is PP2A, which has been implicated in autophagy regulation (*Magnaudeix et al., 2013*; *Neisch et al., 2017*; *Yeasmin et al., 2016*) and shown to interact with WIPI2 via a PP2A regulatory subunit, PPP2R1A (*Bakula et al., 2017*). An age-related decrease in phosphatase levels in neural tissues has been implicated in Alzheimer's disease (*Sontag and Sontag, 2014*), and PP2A has been shown to decrease in neural tissues with age (*Veeranna et al., 2011*), suggesting that declining PP2A activity could contribute to the defect we observed.

Our data suggest that successful autophagosome biogenesis is dependent on the localized environment surrounding the isolation membrane. We observed that a given DRG distal tip could contain both productive and stalled events (*Figure 3—figure supplement 1*). We also detected WIPI2B (S395E)-positive and –negative AVs within a given DRG axonal tip (*Figure 8D*). Since the rate of AV biogenesis decreased with age in neurons (*Figure 1*), at least one component critical to the process is likely altered during aging. However, we now propose that rather than a change in protein level, the critical component might become mislocalized during aging. This age-related mislocalization would prevent an isolation membrane from developing into a productive autophagosome, explaining the observed decreased rate of AV biogenesis with age.

In this study, we focused on the effects of aging on neuronal autophagosome biogenesis. However, there are critical steps in the autophagy pathway downstream from initial autophagosome biogenesis, including autophagosome closure, fusion with lysosomes, retrograde transport of autophagosomes and autolysosomes to the soma, and degradation of cargo. Aging also likely affects these later stages of autophagy. For example, lysosomal integrity has been shown to decrease with age in neurons (*Nixon, 2017*). In non-neuronal cells, retrograde transport of autophagosomes appears to decrease with age in primary mouse fibroblasts (*Bejarano et al., 2018*). It will be interesting to investigate how the later stages of autophagy are altered with age in neurons.

Misregulation of autophagy has been implicated in many neurodegenerative diseases and disorders (*Haack et al., 2012*; *Komatsu et al., 2007*; *Komatsu et al., 2006*; *Nixon et al., 2005*; *Saitsu et al., 2013*). While ectopic induction of autophagy has met with some success in attenuating aggregated mutant huntingtin and Tau in neurodegeneration models (*Ravikumar et al., 2002*; *Wang et al., 2009*), our data suggest that targeting the autophagy initiation complex may not be generally effective for treatment of age-related neurodegenerative disease. Rather, modulating

other stages of autophagosome biogenesis, such as dynamic WIPI2 phosphorylation at the isolation membrane, may produce more successful therapies.

# Materials and methods

## Key resources table

| Reagent type (species) or resource | Designation | Source or reference | Identifiers | Additional information |
|---|---|---|---|---|
| Genetic reagent (*Mus musculus*) | GFP-LC3B | RIKEN BioResource Center in Japan; PMID: 14699058 | RRID:IMSR_RBRC00806 | |
| Genetic reagent (*M. musculus*) | C57BL/6J | Jackson Laboratory | Cat # 000664; RRID:IMSR_JAX:000664 | |
| Cell line (*Homo sapiens*) | HEK293 | Gibco (ThermoFisher) | Cat # R70507 | Authenticated by STR profiling; tested negative for mycoplasma |
| Cell line (*H. sapiens*) | HeLa-M | A. Peden (Cambridge Institute for Medical Research) | | Authenticated by STR profiling; tested negative for mycoplasma |
| Transfected construct (*H. sapiens*) | mCherry-ATG13 | This paper | | Subcloned from RRID:Addgene_22875 |
| Transfected construct (*M. musculus*) | mCherry-ATG5 | Addgene; PMID:16645637 | RRID:Addgene_13095 | |
| Transfected construct (*H. sapiens*) | Halo-ATG5 | This paper | | Subcloned from RRID:Addgene_13095 |
| Transfected construct (*H. sapiens*) | SNAP-ATG9 | This paper | | Subcloned from RRID:Addgene_60609 |
| Transfected construct (*H. sapiens*) | Halo-ATG9 | This paper | | Subcloned from RRID:Addgene_60609 |
| Transfected construct (*H. sapiens*) | SNAP-WIPI1A | This paper | | Subcloned from RRID:Addgene_38272 |
| Transfected construct (*M. musculus*) | Halo-DFCP1 | This paper | | Subcloned from RRID:Addgene_38269 |
| Transfected construct (*H. sapiens*) | mScarlet-LC3A | This paper | | Subcloned from RRID:Addgene_73946 |
| Transfected construct (*H. sapiens*) | mCherry-GABARAP | This paper | | Subcloned from RRID:Addgene_73948 |
| Transfected construct (*H. sapiens*) | mCherry-GEC1 (GABARAPL1) | This paper | | Subcloned from RRID:Addgene_73945 |
| Transfected construct (*H. sapiens*) | mCherry-GATE16 (GABARAPL2) | This paper | | Subcloned from RRID:Addgene_73518 |
| Transfected construct (*H. sapiens*) | SNAP-WIPI2B WT | This paper | | Subcloned from GFP-WIPI2B, PMID: 24954904 |
| Transfected construct (*H. sapiens*) | Halo-WIPI2B WT | This paper | | Subcloned from GFP-WIPI2B, PMID: 24954904 |
| Transfected construct (*H. sapiens*) | SNAP-WIPI2B (FTTG) | This paper | | Generated via quick change from SNAP-WIPI2B WT |

*Continued on next page*

Continued

| Reagent type (species) or resource | Designation | Source or reference | Identifiers | Additional information |
|---|---|---|---|---|
| Transfected construct (*H. sapiens*) | Halo-WIPI2B (FTTG) | This paper | | Generated via quick change from Halo-WIPI2B WT |
| Transfected construct (*H. sapiens*) | SNAP-WIPI2B (R108E) | This paper | | Generated via quick change from SNAP-WIPI2B WT |
| Transfected construct (*H. sapiens*) | Halo-WIPI2B (R108E) | This paper | | Generated via quick change from Halo-WIPI2B WT |
| Transfected construct (*H. sapiens*) | SNAP-WIPI2B (S395A) | This paper | | Generated via quick change from SNAP-WIPI2B WT |
| Transfected construct (*H. sapiens*) | Halo-WIPI2B (S395A) | This paper | | Generated via quick change from Halo-WIPI2B WT |
| Transfected construct (*H. sapiens*) | SNAP-WIPI2B (S395E) | This paper | | Generated via quick change from SNAP-WIPI2B WT |
| Transfected construct (*H. sapiens*) | Halo-WIPI2B (S395E) | This paper | | Generated via quick change from Halo-WIPI2B WT |
| Antibody | Anti-Actin, Mouse Polyclonal | EMD Millipore | Cat # MAB1501; RRID:AB_2223041 | WB (1:3000) |
| Antibody | Anti-ATG10, Rabbit Polyclonal | Novus | Cat # NBP2-38524 | WB (1:100) |
| Antibody | Anti-ATG12, Rabbit Polyclonal | Abcam | Cat # ab155589 | ICC (1:50) |
| Antibody | Anti-ATG13, Rabbit Polyclonal | Abcam | Cat # ab105392; RRID:AB_10892365 | ICC (1:50), IHC (1:100) |
| Antibody | Anti-ATG14, Rabbit Monoclonal | Cell Signaling Technology | Cat # 96752; RRID:AB_2737056 | WB (1:1000) |
| Antibody | Anti-ATG16L1, Rabbit Monoclonal | Abcam | Cat # ab187671 | WB (1:1000), ICC (1:50) |
| Antibody | Anti-ATG3, Rabbit Monoclonal | Abcam | Cat # ab108251; RRID:AB_10865145 | WB (1:3000), ICC (1:50) |
| Antibody | Anti-ATG5, Rabbit Monoclonal | Abcam | Cat # ab108327; RRID:AB_2650499 | WB (1:1000) |
| Antibody | Anti-ATG5-Alexa647, Rabbit Monoclonal | Abcam | Cat # ab206715 | ICC (1:50 ON) |
| Antibody | Anti-ATG7, Rabbit Monoclonal | Abcam | Cat # ab133528; RRID:AB_2532126 | WB (1:1000), ICC (1:50) |
| Antibody | Anti-ATG9, Armenian Hamster Monoclonal | Abcam | Cat # ab187823 | ICC (1:100), IHC (1:100) |
| Antibody | Anti-ATG9-Alexa488, Rabbit Monoclonal | Abcam | Cat # ab206252 | ICC (1:50 ON) |
| Antibody | Anti-Beclin1 (BECN1), Mouse Monoclonal | Santa Cruz | Cat # sc-48341; RRID:AB_626745 | WB (1:200) |
| Antibody | Anti-GABARAP, Rabbit Monoclonal | Cell Signaling Technology | Cat # 13733; RRID:AB_2798306 | WB (1:1000), ICC (1:50) |
| Antibody | Anti-GAPDH, Mouse Monoclonal | Abcam | Cat # ab9484; RRID:AB_307274 | WB (1:1000) |
| Antibody | Anti-GABARAPL1 (GEC1), Rabbit Monoclonal | Cell Signaling Technology | Cat # 26632; RRID:AB_2798928 | WB (1:1000), ICC (1:200) |

Continued

| Reagent type (species) or resource | Designation | Source or reference | Identifiers | Additional information |
|---|---|---|---|---|
| Antibody | Anti-GABARAPL2 (GATE16), Rabbit Polyclonal | Abcam | Cat # ab137511 | ICC (1:50) |
| Antibody | Anti-GABARAPL2 (GATE16), Rabbit Monoclonal | Cell Signaling Technology | Cat # 14256; RRID:AB_2798436 | WB (1:1000) |
| Antibody | Anti-GABARAPs, Rabbit Monoclonal | Abcam | Cat # ab109364; RRID:AB_10861928 | ICC (1:100) |
| Antibody | Anti-Halo, Mouse Monoclonal | Promega | Cat # G9211 | WB (1:500), IP (3 μg) |
| Antibody | Anti-HSP90, Rabbit Monoclonal | Cell Signaling Technology | Cat # 4877; RRID:AB_2233307 | WB (1:1000) |
| Antibody | Anti-LC3, Rabbit Polyclonal | Abcam | Cat # ab48394; RRID:AB_881433 | IHC (1:200) |
| Antibody | Anti-LC3A, Rabbit Monoclonal | Cell Signaling Technology | Cat # 4599; RRID:AB_10548192 | WB (1:1000) |
| Antibody | Anti-LC3B, Mouse Monoclonal | Santa Cruz | Cat # sc-376404; RRID:AB_11150489 | WB (1:100), ICC (1:50) |
| Antibody | Anti-MEK1/2, Rabbit Monoclonal | Cell Signaling Technology | Cat # 8727; RRID:AB_10829473 | WB (1:1000) |
| Antibody | Anti-NFH, Chicken Polyclonal | Aves | Cat # NFH; RRID:AB_2313552 | IHC (1:400) |
| Antibody | Anti-p62, Mouse Monoclonal | Abcam | Cat # ab56416; RRID:AB_945626 | WB (1:200) |
| Antibody | Anti-phospho-ATG14(S29), Rabbit Polyclonal | Cell Signaling Technology | Cat # 13155; RRID:AB_2798133 | WB (1:1000) |
| Antibody | Anti-phospho-ULK1(S757), Rabbit Polyclonal | Cell Signaling Technology | Cat # 6888; RRID:AB_10829226 | WB (1:500) |
| Antibody | Anti-phospho-WIPI2(S395), Rabbit Polyclonal | Cell Signaling Technology | Cat # 13571; RRID:AB_2798259 | WB (1:1000) |
| Antibody | Anti-phospho-WIPI2(S395), Rabbit Polyclonal | This paper | # STO 316PB2 | WB (1:500), ICC (1:50) |
| Antibody | Anti-SV2, Mouse Monoclonal | Developmental Studies Hybridoma Bank, University of Iowa | Cat # SV2; RRID:AB_2315387 | IHC (1:100) |
| Antibody | Anti-ULK1, Rabbit Monoclonal | Cell Signaling Technology | Cat # 8054; RRID:AB_11178668 | WB (1:500) |
| Antibody | Anti-V-type H + ATPase, Rabbit Polyclonal | Synaptic Systems | Cat # 109–002; RRID:AB_887696 | WB (1:500) |
| Antibody | Anti-WIPI1, Rabbit Polyclonal | ThermoFisher | Cat # PA5-34973; RRID:AB_2552322 | WB (1:500) |
| Antibody | Anti-WIPI2, Mouse Monoclonal | Abcam | Cat # ab105459; RRID:AB_10860881 | WB (1:4000) |
| Antibody | Anti-WIPI3, Mouse IgM Monoclonal | Santa Cruz | Cat # sc-514194 | WB (1:100) |
| Antibody | Anti-WIPI4, Rabbit Polyclonal | ThermoFisher | Cat # PA5-71803; RRID:AB_2717657 | WB (1:250) |
| Antibody | Anti-Chicken IgY-AlexaFluor405, Goat Polyclonal | Abcam | Cat # ab175674 | IHC (1:200) |
| Antibody | Anti-Mouse IgG-AlexaFluor405, Goat Polyclonal | ThermoFisher | Cat # A31553; RRID:AB_221604 | IHC (1:200) |

*Continued*

| Reagent type (species) or resource | Designation | Source or reference | Identifiers | Additional information |
|---|---|---|---|---|
| Antibody | Anti-Armenian Hamster-AlexaFluor647, Goat Polyclonal | Abcam | Cat # ab173004; RRID:AB_2732023 | IHC (1:200) |
| Antibody | Anti-Rabbit-AlexaFluor488, Goat Polyclonal | ThermoFisher | Cat # A11034; RRID:AB_2576217 | IHC (1:200) |
| Antibody | Anti-Armenian Hamster-AlexaFluor488, Goat Polyclonal | Abcam | Cat # ab173003 | ICC (1:200) |
| Antibody | Anti-Rabbit IgG-AlexaFluor546, Donkey Polyclonal | ThermoFisher | Cat # A10040; RRID:AB_2534016 | ICC (1:200) |
| Antibody | Anti-Mouse IgG-AlexaFluor546, Donkey Polyclonal | ThermoFisher | Cat # A10036; RRID:AB_2534012 | ICC (1:200) |
| Antibody | Anti-Rabbit IgG-AlexaFluor647, Donkey Polyclonal | ThermoFisher | Cat # A31573; RRID:AB_2536183 | ICC (1:200), IHC (1:200) |
| Antibody | Anti-Rabbit IgG-IRDye 800CW, Donkey Polyclonal | LI-COR Biosciences | Cat # 926–32213; RRID:AB_621848 | WB (1:20,000) |
| Antibody | Anti-Mouse IgG-IRDye 800CW, Donkey Polyclonal | LI-COR Biosciences | Cat # 926–32212; RRID:AB_621847 | WB (1:20,000) |
| Antibody | Anti-Mouse IgM-IRDye 800CW, Goat Polyclonal | LI-COR Biosciences | Cat # 926–32280 | WB (1:20,000) |
| Antibody | Anti-Rabbit IgG-IRDye 680RD, Donkey Polyclonal | LI-COR Biosciences | Cat # 926–68073; RRID:AB_10954442 | WB (1:10,000) |
| Antibody | Anti-Mouse IgG Light Chain-AlexaFluor680, Goat Polyclonal | Jackson Immuno Research Labs | Cat # 115-625-174; RRID:AB_2338937 | WB (1:20,000) |
| Sequence-based reagent | ON-TARGET plus SMARTpool Mouse Wipi2 siRNA | Dharmacon | Cat # L-057690–01 | Proprietary sequence |
| Sequence-based reagent | ON-TARGET plus non-targeting siRNA | Dharmacon | Cat # D-001810–01 | Proprietary sequence |
| Chemical compound, drug | SNAP-Cell 647-SiR | New England Biolabs | Cat # S9102S | |
| Chemical compound, drug | SNAP-Cell TMR-Star | New England Biolabs | Cat # S9105S | |
| Chemical compound, drug | SNAP-Cell 430 | New England Biolabs | Cat # S9109S | |
| Chemical compound, drug | HaloTag TMR | Promega | Cat # G8251 | |
| Chemical compound, drug | silicon-rhodamine-Halo | K. Johnsson, École Polytechnique Federale de Lausanne | | |
| Chemical compound, drug | JF646-Halo | Luke Lavis, Janelia Farms (HHMI) | | |
| Chemical compound, drug | α-Bungarotoxin-tetramethylrhodamine | Sigma | T0195 | |
| Software, algorithm | Volocity | PerkinElmer | | |
| Software, algorithm | FIJI | PMID: 22743772 | | |
| Software, algorithm | Prism 6, Prism 8 | GraphPad | | |
| Software, algorithm | Adobe Illustrator CS4 | Adobe Systems | | |

## Reagents

GFP-LC3B transgenic mice (strain: B6.Cg-Tg(CAG-EGFP/LC3)53Nmi/NmiRbrc) were generated by N. Mizushima (Tokyo Medical and Dental University, Tokyo, Japan; *Mizushima et al., 2004*) and obtained from RIKEN BioResource Center in Japan. These mice were bred with C57BL/6J mice obtained from The Jackson Laboratory. Hemizygous and wild type littermates were used in experiments. Constructs used include: mCherry-ATG13 (subcloned from Addgene 22875), mCherry-ATG5 (Addgene 13095), Halo-ATG5 (subcloned from Addgene 13095), SNAP-ATG9 and Halo-ATG9 (subcloned from Addgene 60609), SNAP-WIPI1A (subcloned from Addgene 38272), Halo-DFCP1 (subcloned from Addgene 38269), mSarlet-LC3A (subcloned from Addgene 73946), mCherry-GABARAP (subcloned from Addgene 73948), mCherry-GEC1 (GABARAPL1, subcloned from Addgene 73945), and mCherry-GATE16 (GABARAPL2, subcloned from Addgene 73518). SNAP-WIPI2B and Halo-WIPI2B were subcloned from GFP-WIPI2B (*Dooley et al., 2014*). SNAP- and Halo- WIPI2B (FTTG), WIPI2B(R108E), WIPI2B(S395A), and WIPI2B(S395E) were generated via quick change and subcloned into original plasmids. The SNAP backbone was originally obtained from New England Biolabs (NEB), and the Halo backbone was originally obtained from Promega.

## Primary neuron culture

DRG neurons were isolated as previously described (*Perlson et al., 2009*) and cultured in F-12 Ham's media (Invitrogen) with 10% heat-inactivated fetal bovine serum, 100 U/mL penicillin, and 100 µg/mL streptomycin. For live-cell microscopy, DRGs were isolated from P21-28 (young), P90-120 (young adult), P480-540 (aged), or P730-760 (advanced aged) mice and plated on glass-bottomed dishes (MatTek Corporation) and maintained for 2 days at 37°C in a 5% $CO_2$ incubator. Prior to plating, neurons were transfected with a maximum of 0.6 µg total plasmid DNA using a Nucleofector (Lonza) using the manufacturer's instructions. Relevant siRNA was co-transfected with plasmid DNA (25 pmol ON-TARGET plus SMARTpool Wipi2 siRNA, L-057690–01 from Dharmacon). For imaging experiments with siRNA, control neurons were transfected with 30 pmol Cy5-labeled non-targeting siRNA (Dharmacon) per dish and experimental neurons were co-transfected with 5 pmol Cy5-labeled non-targeting siRNA (Dharmacon) to identify which neurons received siRNA. For biochemistry siRNA experiments, control neurons were transfected with 25 pmol ON-TARGET plus non-targeting siRNA (D-001810–01 from Dharmacon). Microscopy was performed in low fluorescence nutrient media (Hibernate A, BrainBits) supplemented with 2% B27 and 2 mM GlutaMAX. For nucleofected constructs that yielded Halo- or SNAP-tagged proteins, DRG neurons were incubated with 100 nM of the appropriate Halo or SNAP ligand (SNAP-Cell 647-SiR, SNAP-Cell TMR-Star, or SNAP-Cell 430 from NEB; HaloTag TMR Ligand from Promega, silicon-rhodamine-Halo ligand from K. Johnsson, École Polytechnique Federale de Lausanne, Lausanne, Switzerland, or JF646-Halo ligand from Luke Levis, Janelia Farms, HHMI) for at least 30 min at 37°C in a 5% $CO_2$ incubator. After incubation, DRGs were washed three times with complete equilibrated F-12 media, with the final wash remaining on the DRGs for at least 15 min at 37°C in a 5% $CO_2$ incubator.

Mice of either sex within the indicated postnatal range (1 month, 3 months, 16–17 months or 24 months) were euthanized prior to dissection. All animal protocols were approved by the Institutional Animal Care and Use Committee at the University of Pennsylvania.

## Live-Cell imaging and image analysis

Microscopy was performed on a spinning-disk confocal (UltraVIEW VoX; PerkinElmer) microscope (Eclipse Ti; Nikon) with an Apochromat 100x, 1.49 NA oil immersion objective (Nikon) at 37°C in an environmental chamber. The Perfect Focus System was used to maintain Z position during time-lapse acquisition. Digital micrographs were acquired with an EM charge-coupled device camera (C9100; Hammamatsu Photonics) using Volocity software (PerkinElmer). Time-lapse videos were acquired for 10 min with a frame every 3 s to capture autophagosome biogenesis. Multiple channels were acquired consecutively, with the green (488 nm) channel captured first, followed by red (561 nm), far-red (640 nm), and blue (405 nm). DRGs were selected for imaging based on morphological criteria and low expression of transfected constructs. To minimize artifacts from overexpression, neurons within a narrow range of low fluorescence intensity were chosen for imaging, ensuring the analyzed neurons expressed low levels of the ectopic tagged proteins.

Time-lapse micrographs were analyzed with FIJI (*Schindelin et al., 2012*). 'Stalled' biogenesis events were defined as mCherry-ATG13, mCherry-ATG5, or Halo-ATG5 puncta that remained visible for at least 5 min. 'Productive' biogenesis events were defined as mCherry-ATG13, mCherry-ATG5, or Halo-ATG5 puncta that persisted for less than 5 min and recruited GFP-LC3B.

## Biochemistry

Brains or DRGs of non-transgenic mice were dissected and subsequently homogenized and lysed. Brains were homogenized individually in RIPA buffer [50 mM NaPO4, 150 mM NaCl, 1% Triton X-100, 0.5% deoxycholate, 0.1% SDS, 1x complete protease inhibitor mixture (Roche), and 1x Halt protease and phosphatase inhibitor cocktail (Thermo)]. DRGs were homogenized in RIPA buffer with a 1.5 mL pestle. Homogenized samples were lysed for 30 min on ice. For the siRNA and overexpression controls, isolated DRGs were plated at 120,000 neurons per 35 mm dish as described above for 2 DIV. For the Halo-WIPI2B overexpression controls, where indicated, neurons were treated with 100 nM BaflomycinA1 (BafA) for 4 hr prior to lysis. Neurons were washed with PBS (50 mM NaPO$_4$, 150 mM NaCl, pH 7.4) and then lysed as above.

Samples were centrifuged at 17,000 x g at 4°C for 15 min. Total protein in each lysate was determined by BCA assay (ThermoFisher Scientific) so that equal amounts of protein were loaded into each lane. All supernatants were analyzed by SDS-PAGE, transferred onto FL PVDF membrane, and visualized with fluorescent secondary antibodies (Li-Cor) on an Odyssey CLx imaging system (Li-Cor). The specificity of relevant antibodies was confirmed by immunocytochemistry as described below or by immunoblot. See Table for antibodies used.

All western blots were analyzed with Image Studio (Li-Cor). Total protein was used as a loading control to control for differences in sample loading. The normalization factor is listed below each blot as a percent.

For brain lysate cytosolic and membrane fractions, brains were dissected from non-transgenic mice and subsequently homogenized and lysed. Brains were homogenized individually in Motility Assay Buffer (MAB) [10 mM PIPES, 50 mM K-Acetate, 4 mM MgCl2, 1 mM EGTA, 2 mM PMSF, 210 µM leupeptin, 1.5 µM pepstatin-A, 52.8 µM N-p-Tosyl-L-arginine methyl ester, 20 mM DTT, and 1x Halt protease and phosphatase inhibitor cocktail (Thermo)]. The homogenate was spun at 17,000 x g for 30 min at 4°C. The resultant supernatant was then spun 95,000 x g for 20 min at 4°C. The supernatant was the cytosolic fraction, and the pellet was resuspended in an equal volume of MAB.

## Cell culture and immunoprecipitation

HeLa-M (A. Peden, Cambridge Institute for Medical Research) and HEK293 (ThermoFisher, R70507) cells were cultured in complete medium (DMEM supplemented with 10 fetal bovine serum and 2 mM GlutaMAX). Lipofectamine 2000 (Invitrogen) and FuGENE (Promega) were used to transiently transfect HEK293 and HeLa-M cells, respectively. Immunoprecipitation was performed 24 hr after transfection. Cells were permeabilized with TNTE buffer (20 mM Tris HCl, pH 7.5, 150 mM NaCl, 1% triton TX-100, 5 mM EDTA, and 1X Halt Phosphatase and Protease Inhibitor, ThermoFisher). The HeLa-M lysates were immunoprecipitated with 3 µg mouse monoclonal Anti-Halo (Promega, G9211) and Dynabeads Protein G (Invitrogen). The immunoprecipitated sample was cleaved from the Dynabeads by boiling in Orange Protein Loading Buffer (Li-Cor). GFP-labeled proteins ectopically expressed in HEK293 cells were immunoprecipitated with GFP-Trap beads. The HEK293 cells were routinely tested for mycoplasma and authenticated by STR (short tandem repeat) profiling by The Francis Crick Cell Services. HEK293 cells were used because they are of human origin, fast growing, easy to transfect, and express ectopic proteins without high toxicity. The HeLa cells were routinely tested for mycoplasma using the MycoAlert detection kit (Lonza, LT07) and authenticated by STR profiling using the GenePrint 10 system (Promega, B9510) at the University of Pennsylvania Perelman School of Medicine DNA Sequencing Facility.

## Quantitative Real-Time PCR (qPCR) cDNA isolation from whole brain

Whole brains were collected from euthanized non-transgenic 3-month-old or 16–17 month-old mice and immediately frozen at −80°C. Brains were homogenized in 2 mL TRIzol reagent (ThermoFisher Scientific, 15596018). 2 mL TRIzol reagent and 800 µL chloroform were added after homogenization. Solution was vortexed for 15 s, incubated at room temperature for 5 min, and centrifuged at 12,000

x g for 15 min at 4°C. Clear aqueous phase was mixed with one volume of 200 proof ethanol. Mixture was transferred to Zymo Quick-RNA miniprep kit (Zymo Research, R1057). Total RNA was immediately transferred to Polytract mRNA Isolation System III (Promega, Z5310) to isolate mRNA. Isolated mRNA was immediately transformed into cDNA using M-MuLV Reverse Transcriptase (New England Biolabs (NEB), M0253L; other NEB reagents: S1330S, M0314S). Nucleic acid concentration and purity was monitored throughout isolation.

## qPCR

10 ng total cDNA was added to a 50 µL qPCR reaction with Luna Universal qPCR Master Mix (NEB, M3003G). Each biological sample was loaded in triplicate into qPCR plate. All biological samples for each gene tested were loaded into a single qPCR plate (Phenix Research Products, MPC-3425 and LMT-RT2), with a reference gene loaded into the same qPCR plate. All primers were initially identified through Primer Bank (https://pga.mgh.harvard.edu/primerbank/index.html) (*Spandidos et al., 2010*; *Spandidos et al., 2008*; *Wang, 2003*; *Wang et al., 2012*). Primers were optimized to have melting temperatures at 62°C and tested to ensure appropriate dynamic range. Final qPCR primers used were:

*Pgk1* Fwd (5'-ATGTCGCTTTCCAACAAGCTGACTTTGGAC),
*Pgk1* Rev (5'-GGACTTGGCTCCATTGTCCAAGCAGAATTTG),
*Rplp0* Fwd (5'-GGGCATCACCACGAAAATCTCCAGAGG),
*Rplp0* Rev (5'-CTGCCGTTGTCAAACACCTGCTGG),
*Ulk1* Fwd (5'-GCAAGTTCGAGTTCTCTCGCAAGGACC),
*Ulk1* Rev (5'-CCACGATGTTTTCGTGCTTTAGTTCCTTCAGG),
*Wipi1* Fwd (5'-GCTGCTTCTCTTTCAACCAAGACTGCACATC),
*Wipi1* Rev (5'-CACGTCAGGGATTTCATTGCTTCCATGGAC),
*Wipi2* Fwd (5'-CCAGGATAACACGTCCCTAGCTGTTGG),
*Wipi2* Rev (5'-CTCTCCACAATGCAGACATCTTCAGTGTCAG),
*Wdr45b* (*Wipi3*) Fwd (5'-CGGGTGTTTTGCATGTGGAATGGAAAATGG),
*Wdr45b* (*Wipi3*) Rev (5'-CAGATCATCACTTTGTTGGGAGGGTATTTCGG),
*Wdr45* (*Wipi4*) Fwd (5'-GCGCCATTCACTATCAATGCACATCAGAGTG),
*Wdr45* (*Wipi4*) Rev (5'-GGAGGAGTCGTGGCTGAAGTTAATGCAG),
*Map1lc3b* (*Lc3b*) Fwd (5'-CCCAGTGATTATAGAGCGATACAAGGGGGAG),
*Map1lc3b* (*Lc3b*) Rev (5'-CTGCAAGCGCCGTCTGATTATCTTGATGAG),
*Atg5* Fwd (5'-GGCACACCCCTGAAATGGCATTATCC),
*Atg5* Rev (5'-CCTCAACCGCATCCTTGGATGGAC),
*Atg2a* Fwd (5'-CTATCTGTTCCCAGGTGAACGGAGTGG), and
*Atg2a* Rev (5'-CTGGATGCAGCTGTGTCACGATGG).

qPCR was performed on a QuantStudio 3 Real-Time PCR System (ThermoFisher Scientific) controlled by QuantStudio Design and Analysis Software (ThermoFisher Scientific). Normalized target gene expression level is 2ΔΔCt for each gene relative to the indicated reference gene.

## Immunofluorescence

### Immunocytochemistry

DRGs were isolated, plated, and cultured as described above. At 2 DIV, DRGs were fixed in prewarmed 4% paraformaldehyde (Affymetrix) with 4% sucrose in PBS for 8 min at room temperature. Neurons were washed twice with 1X PBS, then incubated in ice-cold 100% methanol at −20°C for 8 min. Neurons were washed twice with 1X PBS, then incubated in detergent-free Cell Block (1X PBS with 1% BSA and 5% normal goat serum) for one hour at room temperature. DRGs were then incubated in Cell Block containing primary antibodies for one hour at room temperature. After three 5-min washes in 1X PBS, DRGs were incubated in Cell Block containing secondary antibodies for one hour at room temperature. For primary antibodies conjugated to fluorophores, DRGs were incubated in Cell Block containing conjugated primary antibodies overnight at 4°C. DRGs were washed three additional times in 1X PBS, then once with ddH₂O. DRGs were then mounted in Prolong Gold, cured overnight in the dark at room temperature, and assessed by spinning disk confocal microscopy. See Table for antibodies used.

## Immunohistochemistry

Extensor digitorum longus (EDL) muscles were dissected from 3-month-old or 16–17 month old non-transgenic mice. EDL muscles were subsequently immersion-fixed in 2% paraformaldehyde (Affymetrix) for 12 min at room temperature. After three washes in 1X PBS, EDL muscles were incubated in 10 µg/mL α-Bungarotoxin-tetramethylrhodamine (TMR-α-Btx, Sigma T0195) for 15 min at room temperature. After three 10-min washes in 1X PBS, EDL muscles were incubated in ice-cold 100% methanol for 5 min at −20°C. EDL muscles were rinsed in 1X PBS, followed by three 10-min washes in 1X PBS. EDL muscles were then incubated in detergent-free Block (1X PBS with 2% BSA) for 2 hours at room temperature. EDLs were then incubated in primary antibodies in detergent-free block overnight at room temperature. After three 10-min washes in detergent-free Block, EDLs were incubated in secondary antibodies in detergent-free Block for four hours at room temperature. EDLs were washed three times for 10 min per wash in 1X PBS. EDLs were mounted in VectaShield (Vector Labs) and assessed by spinning disk confocal microscopy. See Table for antibodies used.

## Electron microscopy

DRGs from non-transgenic mice were isolated as above and plated as spot cultures on glass-bottomed dishes (MatTek Corporation) and maintained for 2 days at 37°C in a 5% $CO_2$ incubator. DRGs were fixed with 2.5% glutaraldehyde, 2.0% paraformaldehyde in 0.1M sodium cacodylate buffer, pH 7.4, overnight at 4°C. For NMJs, non-transgenic mice were euthanized and subsequently perfused with 2.5% glutaraldehyde, 2.0% paraformaldehyde in 1X PBS. EDL muscles were dissected and post-fixed in 2.5% glutaraldehyde, 2.0% paraformaldehyde in 1X PBS overnight at 4°C. EDL muscles were further post-fixed in 2.5% glutaraldehyde, 2.0% paraformaldehyde in 0.1M sodium cacodylate buffer, pH 7.4. Fixed DRGs and NMJs were then transferred to the Electron Microscopy Resource Laboratory at the University of Pennsylvania, where all subsequent steps were performed. After subsequent buffer washes, the samples were post-fixed in 2.0% osmium tetroxide for 1 hr at room temperature and then washed again in buffer, followed by $dH_2O$. After dehydration through a graded ethanol series, the tissue was infiltrated and embedded in EMbed-812 (Electron Microscopy Sciences, Fort Washington, PA). Thin sections were stained with lead citrate and examined with a JEOL 1010 electron microscope fitted with a Hamamatsu digital camera and AMT Advantage image capture software. Regions between DRG cell body densities with maximum neurite invasion were chosen for imaging.

## Additional methods

All image analysis was performed on raw data. Images were prepared in FIJI (*Schindelin et al., 2012*); contrast and brightness were adjusted equally to all images within a series. Figures were assembled in Adobe Illustrator. Prism 6 (GraphPad) was used to plot graphs and perform statistical tests. Prism 8 (GraphPad) was used to plot graphs and perform statistical tests for *Figure 4G–I*. Statistical tests are indicated in the text and figure legends.

To quantify AV biogenesis, GFP-LC3B puncta were tracked manually using FIJI. An AV biogenesis event was defined as the de novo appearance of a GFP-LC3B punctum based on changes in fluorescence intensity over time. For GFP-LC3B puncta that were present at the start of the time-lapse series, only those puncta that increased in fluorescence intensity and/or area with time were counted as AV biogenesis events.

## Additional information

### Funding

| Funder | Grant reference number | Author |
| --- | --- | --- |
| National Institutes of Health | R37 NS060698 | Erika LF Holzbaur<br>Pallavi P Gopal<br>Andrea KH Stavoe |
| Cancer Research UK | FC001187 | Andrea Gubas<br>Sharon Tooze |

| Medical Research Council | FC001187 | Andrea Gubas<br>Sharon Tooze |
|---|---|---|
| Wellcome Trust | FC001187 | Andrea Gubas<br>Sharon Tooze |
| National Institutes of Health | F32 NS100348-01 | Andrea KH Stavoe<br>Erika LF Holzbaur |
| National Institutes of Health | K99 NS109286-01 | Andrea KH Stavoe<br>Erika LF Holzbaur |

The funders had no role in study design, data collection and interpretation, or the decision to submit the work for publication.

### Author contributions
Andrea KH Stavoe, Conceptualization, Resources, Data curation, Formal analysis, Funding acquisition, Validation, Investigation, Visualization, Methodology, Writing—original draft, Project administration, Writing—review and editing; Pallavi P Gopal, Investigation, Visualization, Writing—review and editing; Andrea Gubas, Investigation, Methodology, Writing—review and editing; Sharon A Tooze, Conceptualization, Supervision, Writing—review and editing; Erika LF Holzbaur, Conceptualization, Supervision, Funding acquisition, Writing—original draft, Project administration, Writing—review and editing

### Author ORCIDs
Andrea KH Stavoe ⓘ https://orcid.org/0000-0002-4073-4565
Erika LF Holzbaur ⓘ https://orcid.org/0000-0001-5389-4114

### Ethics
Animal experimentation: The work in this study was performed in accordance with the Guide for the Care and Use of Laboratory Animals of the National Institutes of Health. All animal protocols were approved by the Institutional Animal Care and Use Committee at the University of Pennsylvania. All animals were euthanized prior to tissue harvest.

### Decision letter and Author response
Decision letter https://doi.org/10.7554/eLife.44219.030
Author response https://doi.org/10.7554/eLife.44219.031

## Additional files

### Supplementary files
• Transparent reporting form
DOI: https://doi.org/10.7554/eLife.44219.028

### Data availability
All data generated or analyzed during this study are included in the manuscript and supporting files.

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
