## [Decision Letter]

[Editors’ note: a previous version of this study was rejected after peer review, but the authors submitted for reconsideration. The first decision letter after peer review is shown below.]

Thank you for submitting your work entitled "Expression of WIPI2B counteracts age-related decline in autophagosome biogenesis in neurons" for consideration by *eLife*. Your article has been reviewed by three peer reviewers, and the evaluation has been overseen by a Reviewing Editor and a Senior Editor. The reviewers have opted to remain anonymous.

Our decision has been reached after consultation between the reviewers. Based on these discussions and the individual reviews below, we regret to inform you that your work will not be considered further for publication in *eLife*.

While the reviewers appreciate the potential new mechanism of a defect in autophagosome biogenesis in aged neurons, the data presented here are rather preliminary and some are confusing. The major comments that you will find below are all important to make this study more complete, but it would take more than the usual 2-month revision period that is allocated for submission of a revised manuscript.

Reviewer #1:

Stavoe et al. report that autophagosome biogenesis is abrogated in neurons from aged mice when compared to young mice. In aged neurons, stalled Atg-13-positive but LC3B-negative structures accumulate. The amount of the WIPI2B protein is smaller in neurons of aged mice compared to young mice, and knockdown of WIPI2B or WIPI1A in young neurons sufficiently induces formation of stalled structures. Furthermore, overexpression of WIPI2B or WIPI1A in neurons from aged mice rescues the failure of autophagosome biogenesis. These results suggest that a decrease in WIPI2B is the cause of defects in autophagosome biogenesis in neurons in aged mice. Although these findings are timely and potentially important, it is unclear what the stalled structures in neurons of aged mice are and whether autophagy flux is indeed affected.

Major comments:

1) The nature of stalled Atg-13-positive LC3B-negative autophagic structures is unclear. Are they autophagosome precursors, autophagosomes, or protein aggregates? In Figure 4, the authors show the presence of multilamellar bodies in neurons from aged mice, but whether these indeed represent stalled isolation membranes is not clear. It is essential to perform CLEM (or immuno-EM) to characterize the stalled structures. Are these undigested structures positive for multiple ATGs and LC3-family proteins except for LC3B as suggested in Figure5C-G?

2) The in vivo data of stalled structures is also weak. CLEM or immune EM would be required for characterization of Atg9- and Atg13-positive structures in aged mice in Figure 3F. Antigen specificity of anti-Atg13 and anti-Atg9 antibodies used in immunohistochemistry of NMJs should also be checked.

3) The authors do not investigate whether autophagy flux is indeed affected in neurons from aged mice. The authors should determine the protein and mRNA levels of p62, an autophagic substrate, in both brain and DRG lysates in Figure S6A and S6B, and also measure autophagic flux (e.g., lysosome-dependent LC3 degradation) in primary DRG neurons from young and aged mice.

4) It was previously reported that WIPI2B interacts with ATG16L1 and recruit the ATG12-5-16L1 complex to autophagic membranes (Dooley et al., 2014). However, the present study shows that ATG5 and ATG16L1 can be recruited to autophagic membranes in WIPI2-depleted neurons. This apparent inconsistency needs to be clarified. Furthermore, the authors show that ATG16L1-binding activity of WIPI2B is required for autophagy restoration in aged neurons (Figure 7C), whereas ectopically expression of WIPI1A, which cannot bind to ATG16L1 (Dooley et al., 2014), could restore autophagy (Figure 7F). This discrepancy should also be explained.

5) Why does only LC3B among the LC3/GABARAP family proteins fail to localize to autophagic structures in aged neurons (Figure 5)? It is expected that, if the ATG5-12-16L1 complex is recruited, this complex can recruit LC3B (Fujita et al., 2008a).

6) The authors claim that the WIPI2 protein level is decreased in neurons in aged mice, but the data are rather weak. In Figure 6A, the amount of WIPI2 seems not much different between young and aged DRG. Data from more than three mice should be shown and quantified. Also in Figure S6A, the amount of ATG proteins should be quantified and statistically analyzed. Inclusion of other key ATGs is required to show that WIPI2B is specifically depleted neurons from aged mice.

7) The role of WIPI1 has been controversial (Polson et al., 2010, Behrends et al., 2010, DeJesus et al., PMID:27351204). To convince that WIPI1A is important in neurons in Figure 6E, rescue experiments using siRNA-resistant WT-WIPI1A, WIPI1A (FTTG) and WT-WIPI2B should be performed. Is the phenotype of WIPI1A-knockdown cells similar to that of WIPI2B-knockdown cells? Do stalled structures appear in WIPI2B- as well as WIPI1A-knockdown cells from young mice?

8) In Figure 7, protein levels of overexpressed WIPI2B and WIPI1A should be shown. Are they similar to those in neurons from young mice? Is autophagy flux improved upon WIPI2B or WIPI1A overexpression?

Reviewer #2:

Stavoe and Holzbaur ask if autophagsome biogenesis is altered during aging or the development neurodegenerative disease. Using a model system of in vitro cultures of primary DRG neurons from young or aged mice they show initial biogenesis events proceed normally but that in aged neurons autophagosomes do not acquire LC3B. In addition they appear to retain ATG13, and 14, and also are positive for the ATG12-5-16L1 complex. They furthermore show differences in Atg9 dissociation from the forming autophagosomes, leading to their hypothesis that in aged neurons autophagosome formation stalls at the stage of acquiring LC3B. Surprisingly they show that LC3A and the GABARAP familes when overexpressed can associate with the stalled autophagosomes but does not rescue. To understand the cause of the stalling ask if the ATG proteins have different mRNA or protein and discover WIPI2 levels decrease but not WIPI1. They show that both WIPI1 and WIPI2 can rescue of LC3B recruitment. Finding a decrease in WIPI2 in aged neurons they show WIPI2B can rescue LC3 recruitment in aged neurons dependent upon the lipid binding motif in WIPI2B, but not with a mutant unable to bind ATG16L1. Interestingly, WIPI1 can also rescue aged neurons.

The manuscript is well written but the data is at times confusing and is largely correlative. The hypotheses are based on pre-existing knowledge and is lacking any new mechanistic insight in particular regarding the WIPI proteins despite some obvious questions coming from their data.

Major points:

1) The definition of a stalled autophagosome gets confusing. The definition seems to be lack of accumulation of LC3B but given the data on LC3A, and the GABARAPs this should be revisited. Do the autophagosome in the old mice is there endogenous GARARAP on the LC3B negative autophagosomes? How do the authors explain the differences between the LC3B and LC3A, and the GABARAPs regarding their association with the stalled autophagosomes?

2) What is the correlation between the decrease in WIPI2 levels and the lack of recruitment of LC3B. If ATG12-5-16L1 are present LC3B should be recruited. Furthermore, the ATG16L1 binding mutant of WIPI2b cannot rescue aged neurons suggesting that the defect is the recruitment of the Atg12-5-16L1 complex but the authors show the ATG12-5-16L1 is present.

3) What is the molecular basis for the requirement for both WIPI1 and WIPI2? In particular as the WIPI1 levels do not change over age? In this regard the experiment test the effect of "restore WIPI1 levels" in Figure 7F is confusing.

4) The knockdown experiments are not conclusive for WIPI1 as the authors themselves point out only 37% (WIPI1) knockdown is achieved. How do they know the cells they study are depleted of WIPI1?

Reviewer #3:

This manuscript by Stavoe et al. demonstrated that aging influenced the biogenesis of constitutive autophagosomes at the elongation/closure stage, but not at nucleation/ initiation stage. In aged neurons, the aberrant autophagic structures (stalled AVs) were shown to be ATG13-positive, but could not form LC3-positive autophagosomes. The defect was suggested due to reduced levels of WIPI2 in aged neurons and overexpression of WIPI2 in these neurons restored autophagosome biogenesis. This is an interesting topic and could be potentially beneficial for therapeutic purpose. However, the quality of the data, especially fluorescent images, is not satisfied and the evidences presented fail to fully support the main conclusion of the work.

Major concerns:

1) The authors showed that the stalled AVs maintained stably levels of mCherry-ATG13/ATG5, without recruitment of LC3B, indicating that early autophagic structures accumulate in aged neurons. However, whether these puncta are autophagic structures is still a question. mCherry is easy to form protein aggregates, especially when expressed at high levels. The formation of mCherry tagged ATG13/ATG5 puncta may be due to decreased degradation capability of proteasome and/or lysosome function in aged neurons. According to Figures 2B, 3A, S1B and S2B, the authors only showed the already exist mCherry-ATG13/ATG5 puncta. Will these puncta eventually disappear? How about newly formed puncta? Can they all become LC3 positive or not? The authors also should perform endogenous antibody staining to comfirm these data.

2) Quantification data should be provided for images in Figures 3E, 3F and S3A-D. Are there more ATG13 puncta accumulated in aged NMJs, compared to young NMJs?

3) The EM results in Figure 4 are not convincing. The multilamellar structures have been demonstrated to belong to lysosomal compartments by multiple literatures. Immuno-EM should be provided to support that these are ATG13 and/or ATG5 positive structures. Quantification data are also needed.

4) Compared to DRG lysates of 3 mo, the decrease of WIPI2 levels in lysates of 16-17 mo is too subtle. Although the difference from brain lysate is obvious, brain lysates are always contaminated with many other types of neural and non-neural cells. If the recruitment of LC3 and autophagosome formation are defective in aged mice, why the LC3-I and II levels are not changed? How about the levels of autophagic substrate p62?

5) Previous study by Dooley et al. showed that WIPI2 links LC3 conjugation to PI3P by recruiting ATG12-5-16L1 complex. The authors suggested that the reduced WIPI2 levels was the reason for impaired LC3 puncta formation. But this could not explain why stalled AVs are all positive for ATG16L1 and ATG5.

6) Is there any functional defect caused by impaired autophagy in aged neurons, eg. defective axonal growth? Can these defects be rescued by overexpression of WIPI2?

[Editors’ note: what now follows is the decision letter after the authors submitted a new version of their paper for consideration.]

Thank you for submitting your article "Expression of WIPI2B counteracts age-related decline in autophagosome biogenesis in neurons" for consideration by *eLife*. Your article has been reviewed by three peer reviewers, and the evaluation has been overseen by a Reviewing Editor and Ivan Dikic as the Senior Editor. The following individuals involved in review of your submission have agreed to reveal their identity: James H Hurley (Reviewer #3).

The reviewers have discussed the reviews with one another and the Reviewing Editor has drafted this decision to help you prepare a revised submission.

Summary:

The authors start with the observation that the rate of autophagosome biogenesis is significantly decreased in neurons from aged mice. Then they show that when autophagosme biogenesis stalls in aged mice neurons, the stalled autophagosomes fail to recruit LC3B, while recruitment of Atg13, DFCP1, and Atg5/Atg12/Atg16 is not affected. Atg9 is retained in the stalled autophagosomes. Inspired by these data, the authors find that overexpression of WIPI2B restores the rate of autophagosome biogenesis in aged mice, and that rescue requires both PI3P-binding and Atg16-binding functions of WIPI2B. Oddly, WIPI2B expression is unaffected. The authors do find, however, that dephosphorylated WIPI2B is required for productive autophagosome biogenesis, and that WIPI2B is dynamically rephosphorylated at the autophagosome to enable the autophagosome to grow.

The last part of the study, concerning the aspect of WIPI2B responsible for the deficit in aging and therefore the mechanism of rescue, is the only part that it is not fully convincing. The question as to the identity of the putative phosphatase is left open. This work advances the field and suggests an interesting avenue for anti-aging interventions. The somewhat less convincing part of the story concerns WIPI2B phosphoregulation.

Essential revisions:

The major concern of all reviewers was the last part of the manuscript and it was generally accepted that WIPI2 phosphorylation section needs extra experimental support. The authors should perform some additional experiments to validate their conclusion that dysfunctional regulation of WIPI2B phosphorylation underlies the failure of autophagosome maturation in aging neurons. They do not need to identify the kinase/phosphatase involved in this regulation, but should strengthen their data as suggested by reviewer 1 (comment 6), reviewer 2 (comments 6-10) and reviewer 3. They should also provide some quantification of the EM data in Figure 1 to support their claims (reviewer 1 comment 1 and reviewer 2 comment 1) and measure autophagic flux upon WIPI2B overexpression. The authors should be able to address these points, as well as the minor comments raised by the three reviewers within two months.

Reviewer #1:

The specific requirement of LC3B recruitment over other ATG8s for autophagosome progression, the presence of ATG9 in stalled autophagosomes, and the nature of the multi-lamellar aberrant vesicles accumulated with age, should at least be further discussed.

1) Figure 1 – rate of AV biogenesis (as measured by LC3B-positive punctae) is decreasing with age, but how does this relate to overall numbers of AVs in DGNs? Multiple studies have suggested increases in Atg8-positive punctae over time (Hansen et al., 2018). It will be important to discuss similarities and differences to published data.

Regarding EM data, the size and frequency of the multilamellar structures in young vs old neurons must be provided to quantitatively support claims.

2) Figure 2 – the conclusion from Figure 2 that there are no changes in elongation of AVs, is not fully supported by the data in Figure 2E (which is out of order, following the role in the biogenesis process). Moreover, Figure 2H is missing time points analyzed for initiation and elongation, and should be added for consistency.

3) Figure 4 – Authors report ATG9 remains in stalled autophagosomes. As WIPI4 is required for ATG9 retrieval, the authors should consider analyzing levels of WIPI4 with age. Moreover, Figure 4B is missing young data (subsection “Stalled AVs recruit autophagosome biogenesis components”).

4) Figure 5 – The authors need to quantify and present statistics for the fraction of stalled events that could recruit ATG12, ATG7, ATG16L1, and ATG3.

Furthermore, the authors interestingly find that all ATG8 members except LC3B are recruited to autophagosomes in old neurons. However, no data are shown towards the experiments by which overexpression of these family members have been tested (subsection “Stalled AVs recruit autophagosome biogenesis components”); this is key to add, including controls to assess levels of overexpression. How do the overall number of other ATG8 proteins change over time? Moreover, knowing that the machinery required for ATG8 recruitment (including WIPI2 and ATG16L1) is the same for all ATG8 proteins, the authors should at the very least discuss or speculate what is special for LC3B among other homologs.

5) Figure 6 – WIPI2B overexpression in old neurons restored autophagosome biogenesis. However, it is not clear whether these autophagosomes are bona fide autophagosomes. The authors should use a lipidation mutant form of LC3B to test the identity of the observed structures, i.e., to assess if they are likely to be autophagosomes versus random aggregates. Flux assay should also be performed to assess whether autophagy is induced.

EM micrographs show accumulation of multilamellar structures in neurons with age, that the authors claim are of autophagic origin. While the reviewer understands that labeling these structures with autophagy markers may be technically challenging, the authors should at least consider analyzing how WIPI2 levels affect their numbers (ideally decreasing them).

6) Figure 7 – The authors hypothesized that the dynamic phosphorylation of WIPI2B is key for the recruitment of LC3B and autophagosome formation. However, evidence to support this is not solid (e.g., evaluation of cells overexpressing the phospho-mimetic mutant, including expression levels, localization etc are missing, and Figure 8A is especially open-ended). One possible experiment the authors could do is to express WIPI2B (S395A) and WIPI2B (S395E) in the presence of siRNA for endogenous WIPI2 to test if these two forms of WIPI2B are recruited to the forming autophagosome in a sequential manner. Importantly, the authors must test their pWIPI2 antibodies specificity using a negative control, e.g., immunoprecipitated WIPI2 phospho-null mutant.

The authors describe that age-dependent defects on autophagosome biogenesis are likely due to aberrant post-translational modification of WIPI2 rather than WIPI2 levels. How can they explain that WIPI2 overexpression then rescues age-dependent defects? This point must be discussed in detail.

Reviewer #2:

1) Although not absolutely required, it would be nice to get some numbers to support the conclusions drawn based on Figure 1 C-J that aged neurons show more aberrant autophagic vesicles.

2) Both the text and the figure legends of several figures (e.g. Figure 2 and 3) fail to mention that quantification of puncta formed is based on live-cell imaging of neurons transfected with fluorescent constructs. The figures should also be clearly labeled with the protein analyzed (e.g. Figure 2B-D should be labeled mCherry-ATG13 instead of ATG13) for the reader to know if one is looking av transfected or endogenous protein. They should also explain (at least in the Materials and methods) how the quantifications (rate of formation) were done.

3) To confirm that the reduced level of autophagosome formation seen with age is real, and not due to reduced level of transfection in aged neurons, the authors should show by western blot that the level of transfection is equal for young and old neurons. Moreover, they should confirm their data using staining for endogenous proteins (LC3).

4) In Figure 4E-F, the authors nicely show the presence of stalled AVs in vivo in old mice, as detected by colocalization of ATG9 and ATG13. They should also show that these structures lack LC3B to confirm their in vitro data.

5) In Figure 6 the authors ask if WIPI2 could be involved in the stalling phenotype of AVs seen in neurons from aged mice. They should first show if WIPI2 is recruited or not to stalled AVs, as they have done for all other autophagy markers. Now, they directly shown the importance of WIPI2 and its binding to PI3P and ATG16L1 in autophagosome biogenesis in primary neurons, which is nice and expected, but the transition would have been better if they could show 6E-H before A-D (at least for WT WIPI2).

6) In Figure 7 they show nicely that de-phosphorylation of WIPI2B at forming AVs seems to be required for recruitment of LC3B and AV biogenesis. However, they do not detect a difference in protein levels of total or phosphorylated WIPI2B with age and therefore propose that changes in cytosol/membrane levels of phosphorylated WIPI2B could change with age. This can be easily tested (and should be done) by a crude cytosol/membrane fractionation of their cell lysates or alternatively by gradient fractionation to see if the l specificity of membrane binding of P-WIPI2B is changed. From the blots in Figure S4 and S5 it seems like the majority of WIPI2B is phosphorylated and that the upper band of the WIPI2 blot correspond to the P-WIPI2 band. Is this correct?

7) Figure 7H; this figure is confusing. The authors claim that it shows a stalled AV where WIPI2B S395E is not recruited, but how do we know ATG5 and WIPI2B S395E are expressed?

8) Figure 8B: the figure suggests a model where WIPI2B de-phosphorylation regulates its membrane recruitment and binding PI3P. To be able to conclude about this, they should do the membrane fractionation experiments suggested above and/or colocalization with PI3P.

9) Furthermore, the model indicates that WIPI2B de-phosphorylation is important for the initial membrane recruitment of LC3B (and AV biogenesis), while WIPI2B re-phosphorylation is required for growth of LC3B positive structures. Their data are however not of sufficient strength to conclude about this and they should consider to modify the model as well as the sentence stating: " Further, we find that the dynamic regulation of WIPI2B phosphorylation at the isolation membrane is integral to autophagosome biogenesis, as our results suggest that dephosphorylated WIPI2B is required for recruitment of LC3B to the isolation membrane, while phosphorylated WIPI2B promotes expansion of the autophagosome".

10) It is puzzling that the WIPI2B S395E mutant is not recruited to the early structures, as there is no difference in recruitment of DFCP1 (showing the presence of PI3P) and ATG5-12-16L1. Is the WIPI2B S395E mutant able to interact with PI3P and ATG16L1?

Reviewer #3:

The data are generally quantified and statistics included (with description of which test used in the figure legends), but the authors should include more information about the software used for quantification of rate of AV formation.

This conclusion that autophagosome formation slows in aged mouse neurons is mainly supported by Figure 1 and Figure S1, which are quite convincing. Could the authors give more details about the definition of the rate of autophagosome biogenesis, i.e. LC3B puncta formation per min or mature autophagosome formation per min?

Figure 2, Figure 3, and part of Figure 5 support the conclusions about the step at which the defect occurs. Figure 4 and 5 show that Atg9 is retained in the stalled autophagosomes, convincingly. These findings are also quite clear. I am curious about the destiny of the stalled autophagosomes. Are they eventually degraded by lysosomes? This should be discussed.

Figure 6 contains the key rescue data. It is interesting that WIPI2B function depends on Atg16-binding, but the Atg16 recruitment is not affected in aged mouse neuron. Please discuss this.

[Editors’ note: further revisions were requested prior to acceptance, as described below.]

Thank you for submitting your article "Expression of WIPI2B counteracts age-related decline in autophagosome biogenesis in neurons" for consideration by *eLife*. Your article has been reviewed by two peer reviewers, and the evaluation has been overseen by a Reviewing Editor and Eve Marder as the Senior Editor. The following individuals involved in review of your submission have agreed to reveal their identity: Anne Simonsen (Reviewer #2).

The reviewers have discussed the reviews with one another and the Reviewing Editor has drafted this decision to help you prepare a revised submission.

Summary:

The authors show that the rate of autophagosome biogenesis is significantly decreased in neurons from aged mice. The authors find that overexpression of WIPI2B restores the rate of autophagosome biogenesis in aged mice, and that rescue requires both PI3P-binding and Atg16-binding functions of WIPI2B. The authors propose that dephosphorylated WIPI2B is required for productive autophagosome biogenesis, and that WIPI2B is dynamically rephosphorylated at the autophagosome to enable the autophagosome to grow. The last part of the study, concerning the aspect of WIPI2B responsible for the deficit in aging and therefore the mechanism of rescue, has been addressed in the revised version. The authors have made significant effort to address the reviewer's comments, including further proving the relevance of WIPI2B phosphorylation in autophagosome biogenesis. Specifically, the authors added an experiment to quantify the membrane/cytosolic ratio of phospho-WIPI2B and found that phospho-WIPI2B is less likely to be associated with the membrane. Overall, the work has been improved based on the initial review. This work advances the field and suggests an interesting avenue for anti-aging interventions and all reviewers were positive about the overall message of the manuscript.

Essential revisions:

Although the authors have convincingly shown that overexpression of WIPI2B or a phospho-null version of this protein in aged neurons restores autophagosome biogenesis, the sequential phosphorylation model of WIPI2B to control autophagosome biogenesis should be more carefully discussed and main messages on dynamics toned down.

1) Overexpression of both wild type or phospho-dead WIPI2(S395A) can rescue autophagosome biogenesis. How do the authors explain that a transition between dephosphorylation and phosphorylation is required for autophagosome biogenesis? The authors found that WIPI2 proteins levels increased with age in DRG neurons (Figure 6—figure supplement 1), but this result is discussed as if there is no change of WIPI2 levels with age. Combined with the observation that phosphorylation of WIPI2 is unchanged over time, this raises the possibility that overexpression of WIPI2 might not automatically lead to increases in phosphorylation (as argued by the authors in a rebuttal response). This brings about new questions, like how much (ie how many fold) is WIPI2 actually overexpressed in neurons compared to the age-associated increase observed, and has increases in WIPI2 phosphorylation (including over time) indeed been verified in this setting?

2) In Figure 5 A-E, the authors used co-localization of ATG9 with ATG5 to identify stalled AV. However, while the authors showed representative pictures in Figure 5—figure supplement 1 that some LC3B homologs co-localize with both ATG9 and ATG5, the authors used ATG5 alone to identify stalled AV in Figure 5 F-J. It is possible that these vesicles may be functional autophagic vesicles with other LC3 homologs instead of LC3B. The authors may consider these as stalled AV as ATG5 stay long on the vesicle, but an alternative explanation may be that autophagic vesicles with other LC3 homologs have a different dynamics compared to that of LC3B.

Figure 2L – The number of ATG5-positive punctae were found to decrease over time, as now directly stated in the paper, but the significance of this result is not discussed at all.

3) Although the authors do not wish to make conclusions about their newly conducted flux assays, the provided diagram (which unfortunately is not accompanied with a figure legend, making it impossible to fully evaluate) may indicate that autophagy is still active in aged DRG neurons. If so, how would the authors explain the remaining autophagy activity with a decreased biogenesis of autophagosome vesicles in aged neurons?

4) Missing data: Data showing ectopic overexpression of different mAtg8s to test LC3B recruitment to stalled AVs were not included, as requested by this reviewer – this makes it impossible to evaluate the full experiment, including the negative data.

Quantification data is missing in Figure 4E-F. Authors should provide how many events they have observed in young and old NMJs.

5) The authors' argument not to cite Berjarano et al., 2018 because the study is not conducted in neurons is not valid. This study directly tested and showed, for the first time, an age-related decline in transport of autophagosomes, a conceptual advance that it relevant to this paper's findings, irrespective of cell type.

---

## [Author Response]

[Editors’ note: the author responses to the first round of peer review follow.]

Thank you for submitting your work entitled "Expression of WIPI2B counteracts age-related decline in autophagosome biogenesis in neurons" for consideration by eLife. Your article has been reviewed by three peer reviewers, and the evaluation has been overseen by a Reviewing Editor and a Senior Editor. The reviewers have opted to remain anonymous.Our decision has been reached after consultation between the reviewers. Based on these discussions and the individual reviews below, we regret to inform you that your work will not be considered further for publication in eLife.While the reviewers appreciate the potential new mechanism of a defect in autophagosome biogenesis in aged neurons, the data presented here are rather preliminary and some are confusing. The major comments that you will find below are all important to make this study more complete, but it would take more than the usual 2-month revision period that is allocated for submission of a revised manuscript.

The three referees who reviewed our initial submission at eLife concluded that the manuscript was timely and potentially important, but significantly more work was required to fully develop the story. We agree, and have spent the last 10 months refining our study, focusing on the new and important biological insights we have gleaned. Therefore, we are happy to once again submit this work to *eLife* for consideration for publication.

However, we are requesting that it be considered as a new submission because the direction that the work took was not always the same as that suggested by the initial reviewers. Many of their suggestions were helpful, but we also profited from in-depth discussions with other scientists in the field. In fact, these discussions led to a very exciting new collaboration with Dr. Sharon Tooze, an autophagy expert who is now a co-author of the study.

Importantly, our understanding of the underlying mechanisms has evolved significantly. While initially we thought the deficit might be due to declining levels of WIPI2, based on reviewer feedback we pursued this question in more detail. Extensive quantitative analysis of RNA and protein levels indicate that aging does not significantly lower levels of WIPI2. Instead, aging causes a local mis-regulation of WIPI2 phosphorylation. These studies in turn have provided new and very timely insights into the role of WIPI2B as a master regulator of autophagosome

formation and growth in neurons, through a targeted phosphorylation that is locally regulated within the axon tip. This direction provided a clearer molecular explanation of the aging-dependent decline in autophagosome formation.

In summary, while the points raised by the review of our initially submitted manuscript were helpful in further developing this work, our additional experimental observations have led to a significantly different submission. We feel that this work is best viewed on its own merits. Thus, we request that this current submission be handled as a new, rather than revised, manuscript.

We also agreed with the reviewers that the specific role of WIPI1 in autophagosome biogenesis is not yet clear so our initial observations on WIPI1 are not included in this new, more focused manuscript. While our unpublished data indicate that WIPI1 is required for autophagosome formation, the underlying mechanisms remain to be fully determined.

While we valued the suggestions to further develop the EM approaches, potentially through immuno-EM or CLEM, we focused instead on a more molecular analysis of autophagosome biogenesis. EM is a valuable but qualitative approach; instead we focused on more quantitative measures of the aging deficit we are reporting.

The reviewers did bring up an interesting question, that while Atg16L1 is recruited successfully to stalled autophagosomes, lipidated LC3B is not incorporated into the stalled structures seen in neurons from aged mice. This observation points out the current limitations in our overall understanding of the autophagosome biogenesis pathway. These results, in conjunction with other recent observations, suggest that LC3B recruitment is not sufficient for AV generation and elongation. In contrast, our new data indicate that dynamic WIPI2B phosphorylation is required, and this is what we have focused on during our revisions.

Finally, we did look in detail to see if flux was altered or autophagosome substrates such as p62 accumulated in neurons from aged mice. We saw no change in the overall LC3I/II ratio in either aged neurons or aged brain, and did not note any changes in p62 with age. However, these are insensitive measures of axonal autophagy, which is compartmentalized in neurons and does not require receptors involved in selective autophagy such as p62. Importantly, we are not studying

a genetic knockout, we are examining the effects of the normal aging process on axonal autophagy. Thus, the downstream effects of aging-related deficits in autophagosome formation are likely to be less marked, and may require specific challenges to the system such as induction of proteotoxic stress, which we hope to test in future studies.

[Editors’ note: the author responses to the review of the new version of their paper follow.]

Essential revisions:The major concern of all reviewers was the last part of the manuscript and it was generally accepted that WIPI2 phosphorylation section needs extra experimental support. The authors should perform some additional experiments to validate their conclusion that dysfunctional regulation of WIPI2B phosphorylation underlies the failure of autophagosome maturation in aging neurons. They do not need to identify the kinase/phosphatase involved in this regulation, but should strengthen their data as suggested by reviewer 1 (comment 6), reviewer 2 (comments 6-10) and reviewer 3. They should also provide some quantification of the EM data in Figure 1 to support their claims (reviewer 1 comment 1 and reviewer 2 comment 1) and measure autophagic flux upon WIPI2B overexpression. The authors should be able to address these points, as well as the minor comments raised by the three reviewers within two months.

We thank the reviewers for their thoughtful discussion of our work. We agree with the overall conclusions – our work describes a novel and potentially important age-dependent deficit in autophagosome formation in primary neurons from aging rodents. Rescue experiments provide strong support for the mechanistic interpretation that there is an age-dependent dysfunction in WIPI2B phosphorylation, but further experiments addressing the role of WIPI2 phosphorylation would strengthen this point. We have followed the specific suggestions provided, and now include data that more firmly establish this point, including the requested information on the specificity of the phospho-specific WIPI2 antibody.

Importantly, we found the insights of Reviewer 2 to be correct: fractionation experiments demonstrate that phosphorylated WIPI2 is significantly less likely to be associated with the membrane fraction than is the total pool of endogenous WIPI2 when examining brain lysates from either young or aging mice. We measured a membrane/cytosolic ratio of 5.8% for P-WIPI2 vs. 12.6% for total WIPI2 for 3-month mice (p=0.0001 by one-way ANOVA) and a membrane/cytosolic ratio of 5.0% for P-WIPI2 vs. 9.7% for total WIPI2 for 16-month mice (p<0.003 by one-way ANOVA). These data are now included in the revised Figure 8B.

To address the other key points, we now provide a quantitative analysis of our EM data. By EM analysis, we find that 80.4% of autophagosomes in the distal tips of primary DRG neurons are morphologically normal, significantly different from the 34.0% that are morphologically normal in neurons from aged mice (n=56 from young neurons, n=153 from aged neurons; p<0.0001). We have added this quantification to the text of the revised manuscript, and also include the relevant graph in Author response image 1. We found no significant difference in AV size by age or AV type (normal vs. multilamellar).

Further, we examined autophagic flux upon WIPI2B overexpression. Using the western blot assay with BaflomycinA, we did not detect a change in autophagic flux with overexpression of wild type WIPI2B. However, since transfection efficiencies are very low for primary neurons, it is not surprising that we could not detect a change in autophagic flux from cultured DRG neurons. See Author response image 2.

**Author response image 2. respfig2:** DRG neurons were harvested from 3-mo- or 16-17-mo-old mice and co-transfected with mCh-ATG5 and the indicated Halo-tagged construct. Neurons from 2 mice were pooled during isolation and then split into two samples and nucleofected and plated separately. After 2 DIV, neurons were treated with 2 uL DMSO or 2 uL Bafilomycin A1 (100 nM final concentration) for 4 hours. Neurons were then lysed and harvested. Equal amounts of total protein were loaded onto a SDS-PAGE gel and assessed by immunoblot. LC3 levels were first normalized to total protein, then the LC3-II/LC3-I ratio was calculated and plotted.</

Reviewer #1:The specific requirement of LC3B recruitment over other ATG8s for autophagosome progression, the presence of ATG9 in stalled autophagosomes, and the nature of the multi-lamellar aberrant vesicles accumulated with age, should at least be further discussed.

We now extend our discussion to note the growing literature indicating that the LC3/GABARAP proteins are not functionally redundant. We also discuss the potential significance of the continued association of ATG9 with stalled autophagosomes and how this may explain the age-dependent accumulation of multi-lamellar vesicles. However, in the absence of further mechanistic insights, we would prefer to keep this speculation brief. The relevant sentences are highlighted in the revised manuscript.

1) Figure 1 – rate of AV biogenesis (as measured by LC3B-positive punctae) is decreasing with age, but how does this relate to overall numbers of AVs in DGNs? Multiple studies have suggested increases in Atg8-positive punctae over time (Hansen et al., 2018). It will be important to discuss similarities and differences to published data.

Our study is the first that we are aware of to directly investigate the effects of aging on autophagosome biogenesis in neurons. Other studies have focused on steady state measures of AV content as a function of aging. It is highly likely that other steps in the autophagy pathway, such as fusion, maturation, and clearance, are also affected by aging, leading to the observed increases in Atg8-positive puncta seen in other model systems. We clarify this point in the revised manuscript.

Regarding EM data, the size and frequency of the multilamellar structures in young vs old neurons must be provided to quantitatively support claims.

As requested, we now provide a quantitative analysis of our EM data. By EM analysis, we find that 80.4% of autophagosomes in the distal tips of primary DRG neurons are morphologically normal, significantly different from the 34.0% that are morphologically normal in neurons from aged mice (n=56 from young neurons, n=153 from aged neurons; p<0.0001). As requested, we also quantified the size of the autophagic vesicles and multilamellar structures and found no change with age or type of structure. See Author response image 3.

**Author response image 3. respfig3:** 

2) Figure 2 – the conclusion from Figure 2 that there are no changes in elongation of AVs, is not fully supported by the data in Figure 2E (which is out of order, following the role in the biogenesis process). Moreover, Figure 2H is missing time points analyzed for initiation and elongation, and should be added for consistency.

The rate of formation of ATG5 puncta per min is modestly decreased in DRG neurons from 24-month old mice as compared to 3-month old mice, but not to 1-month or 16-17-month-old mice, as noted in the figure. We now point this out within the text of the revised manuscript. We have also switched the order of the panels, as requested. Finally, we measured the rates of DFCP1 puncta formation at both the 1-month and 24-month time points, as requested, and found that these rates were not significantly different than the previously measured rates for neurons from mice that were 3- or 16-17-months old. These additional data have been added to the revised Figure 2.

3) Figure 4 – Authors report ATG9 remains in stalled autophagosomes. As WIPI4 is required for ATG9 retrieval, the authors should consider analyzing levels of WIPI4 with age. Moreover, Figure 4B is missing young data (subsection “Stalled AVs recruit autophagosome biogenesis components”).

We were able to find antibodies to WIPI3 and WIPI4 and now include those blots and analysis of all WIPI levels with age in Figure 6—figure supplement 1. We also assessed the mRNA levels of WIPI3 and WIPI4 and include that data in Figure 6—figure supplement 1. A reference to Figure 4D was added for the young ATG9 data.

4) Figure 5 – The authors need to quantify and present statistics for the fraction of stalled events that could recruit ATG12, ATG7, ATG16L1, and ATG3.

We did not observe deficits in the recruitment of any of these factors to either productive or stalled events in neurons from aged mice. We quantified the colocalization of endogenous ATG16L1, ATG5, and ATG9 in fixed neurons from aged mice. Of the puncta that were both ATG5- and ATG9-positive, 100% were also ATG16L1-positive (37 puncta from 35 neurons). There were 30 additional puncta that were ATG5- and ATG16L1-positive puncta that were ATG9-negative.

Furthermore, the authors interestingly find that all ATG8 members except LC3B are recruited to autophagosomes in old neurons. However, no data are shown towards the experiments by which overexpression of these family members have been tested (subsection “Stalled AVs recruit autophagosome biogenesis components”); this is key to add, including controls to assess levels of overexpression. How do the overall number of other ATG8 proteins change over time? Moreover, knowing that the machinery required for ATG8 recruitment (including WIPI2 and ATG16L1) is the same for all ATG8 proteins, the authors should at the very least discuss or speculate what is special for LC3B among other homologs.

We did not observe clear differences in expression levels among the members of the LC3/GABARAP family over time (Figure 6—figure supplement 1). While we appreciate the reviewer’s curiosity, we are very hesitant to speculate about the functional differences among these proteins in the absence of more specific data, although as noted above, we now include a brief discussion of the growing body of literature indicating functional differences among these proteins. However, it is interesting to note that the induced recruitment of LC3B to the nascent autophagosome caused by LC3A overexpression was not sufficient to resolve the stalled event, suggesting that it is not the recruitment or lipidation of LC3B that is the issue, but instead this is a marker for an upstream problem, likely related to the sustained presence of ATG9. We did attempt to address this point by overexpressing each of the constructs in DRGs and assessed the levels by western blot. However, due to technical limitations, the antibodies specific to each mAtg8 were not able to detect either endogenous or overexpressed levels in cultured DRGs. In addition, we select neurons to image within a narrow fluorescence intensity range. As fluorescence intensity is directly correlated to total construct expression, we can be certain that we are imaging and analyzing neurons expressing roughly similar levels of each of the mAtg8s. We have modified the Materials and methods to clarify this.

5) Figure 6 – WIPI2B overexpression in old neurons restored autophagosome biogenesis. However, it is not clear whether these autophagosomes are bona fide autophagosomes. The authors should use a lipidation mutant form of LC3B to test the identity of the observed structures, i.e., to assess if they are likely to be autophagosomes versus random aggregates. Flux assay should also be performed to assess whether autophagy is induced.

**Author response image 4. respfig4:** 

The morphology, rate of formation, and dynamics of GFP-LC3B-positive structures induced in aged neurons by over-expression of WIPI2B are consistent with the conclusion that they are bone fide autophagosomes. At the low levels of GFP-LC3B expression observed in the transgenic mouse (Mizushima, 2004), we do not see random aggregate formation, a likely artifact of high expression levels. As requested, we used a lipidation mutant to confirm that GFP-LC3B-positive structures are not random aggregates.

As requested, flux assays were performed, but this is a less sensitive assay that is insufficient to detect changes at the intracellular level. The flux assays are performed on an entire plate of cultured neurons. Since the transfection rate of neurons is very low, it is not surprising that we did not detect any changes in autophagic flux. See Author response image 2.

EM micrographs show accumulation of multilamellar structures in neurons with age, that the authors claim are of autophagic origin. While the reviewer understands that labeling these structures with autophagy markers may be technically challenging, the authors should at least consider analyzing how WIPI2 levels affect their numbers (ideally decreasing them).

The reviewer is correct, this experiment is too technically challenging to provide quantitative insights given the low transfection efficiencies observed for primary neurons.

6) Figure 7 – The authors hypothesized that the dynamic phosphorylation of WIPI2B is key for the recruitment of LC3B and autophagosome formation. However, evidence to support this is not solid (e.g., evaluation of cells overexpressing the phospho-mimetic mutant, including expression levels, localization etc are missing, and Figure 8A is especially open-ended).

We have assessed the relative expression levels of the WIPI2B wild type, phosphomimetic and nonphosphorylatable constructs; we did not detect differences in expression among the different WIPI2B constructs (Figure 7—figure supplement 1E). Furthermore, when we select neurons to image, we always choose neurons expressing the constructs within the same narrow range of low fluorescence intensity. Since fluorescent signal is directly correlated to expression levels for these constructs, we can be certain that we are assessing neurons expressing similar levels of each of the WIPI2B constructs. We have clarified our explanation of this in the Materials and methods. We have also improved the clarity of our presentation of the data on cellular localization of these constructs, including Figure 8A – now Figure 7H-J in the revised manuscript.

**Author response image 5. respfig5:** 

As addressed above, we have now added additional data characterizing the phospho-specific antibody and the expression levels (Figure 7—figure supplement 1E) and localization of the phospho-mimetic (Figure 7H) and nonphosphorylatable mutant WIPI2B constructs (Figure 7I-J, 8D). We tested both phospho-WIPI2 antibodies against HeLa lysates expressing each of the Halo-WIPI2B constructs (Figure 7—figure supplement 1C). Since Wan et al., 2018 Mol Cell already tested the phospho-WIPI2 antibody from CST (13571), we performed the suggested negative control with the novel antibody (316BP2). In all of these experiments, the phospho-WIPI2 antibodies detected Halo-WIPI2B WT, but neither of the phospho point mutants, which is consistent with what Wan et al., 2018 found. Additionally, we performed the siRNA experiment as suggested, expressing both phospho-WIPI2 constructs simultaneously. We were not able to see sequential recruitment because Halo-WIPI2B(S395A) remained stuck at the puncta. We have included these data in the revised manuscript in Figure 8C.

One possible experiment the authors could do is to express WIPI2B (S395A) and WIPI2B (S395E) in the presence of siRNA for endogenous WIPI2 to test if these two forms of WIPI2B are recruited to the forming autophagosome in a sequential manner.

This was a great suggestion. We could not detect recruitment of the S395A and S395E constructs in a sequential manner because we noted that the S395A construct appears to be retained on the membrane. We then quantified the retention time for each construct and now include the resulting data in Figure 8C.

Importantly, the authors must test their pWIPI2 antibodies specificity using a negative control, e.g., immunoprecipitated WIPI2 phospho-null mutant.

These data have been added to Figure 7—figure supplement 1C-D. Our data are consistent with Wan et al., 2018 Mol Cell.

The authors describe that age-dependent defects on autophagosome biogenesis are likely due to aberrant post-translational modification of WIPI2 rather than WIPI2 levels. How can they explain that WIPI2 overexpression then rescues age-dependent defects? This point must be discussed in detail.

Higher levels of WIP2B expression are likely to increase the cellular concentration of the unphosphorylated form, which is required for successful biogenesis.

Reviewer #2:1) Although not absolutely required, it would be nice to get some numbers to support the conclusions drawn based on Figure 1 C-J that aged neurons show more aberrant autophagic vesicles.

As noted above, we now provide a quantitative analysis of our EM data. By EM analysis, we find that 80.4% of autophagosomes in the distal tips of primary DRG neurons are morphologically normal, significantly different from the 34.0% that are morphologically normal in neurons from aged mice (n=56 from young neurons, n=153 from aged neurons; p<0.0001). See Author response image 1.

2) Both the text and the figure legends of several figures (e.g. Figure 2 and3) fail to mention that quantification of puncta formed is based on live-cell imaging of neurons transfected with fluorescent constructs. The figures should also be clearly labeled with the protein analyzed (e.g. Figure 2B-D should be labeled mCherry-ATG13 instead of ATG13) for the reader to know if one is looking av transfected or endogenous protein. They should also explain (at least in the Materials and methods) how the quantifications (rate of formation) were done.

To quantify AV formation, we examined the GFP-LC3B channel in our time-lapse videos. A biogenesis event was defined as the de novo generation of a LC3B punctum or the growth of a punctum that was present at the start of the video acquisition (growth in terms of visible expansion of the area of the punctum). Since the latter category only applies to the puncta present at the start of the video, the vast majority of the events quantified were de novo GFP-LC3B puncta (that also grew during the time lapse). We have clarified this point in the Materials and methods section of the revised manuscript.

3) To confirm that the reduced level of autophagosome formation seen with age is real, and not due to reduced level of transfection in aged neurons, the authors should show by western blot that the level of transfection is equal for young and old neurons. Moreover, they should confirm their data using staining for endogenous proteins (LC3).

Please see Figure 7—figure supplement 1E for the western blot showing equivalent levels of overexpression of Halo-WIPI2B with age in DRGs. However, western blot analysis only indicates overexpression levels at a whole-plate level. During imaging, we are careful to select neurons within a narrow range of fluorescence intensity. Since fluorescence intensity is directly correlated to construct expression, we are imaging and analyzing neurons expressing roughly equivalent amounts of WIPI2.

We also examined endogenous levels of several autophagy components in whole brain and DRG lysates; see Figure 6—figure supplement 1 for WIPIs, see Figure 6—figure supplement 2 for mAtg8s and other autophagy proteins.

If the reviewer is referring to the autophagy flux assay, we performed this assay on DRG cultures overexpressing the WIPI2B constructs and saw no change in autophagic flux. However, as explained above, primary neuron cultures exhibit low transfection efficiency, so we are not surprised that there was no change in autophagic flux.

4) In Figure 4E-F, the authors nicely show the presence of stalled AVs in vivo in old mice, as detected by colocalization of ATG9 and ATG13. They should also show that these structures lack LC3B to confirm their in vitro data.

Unfortunately, this experiment is not technically possible due to the limited number of independent channels and sources of dependable, IF/ICC-capable antibodies. LC3A and LC3B are especially problematic, as many antibodies that purport to be isoform-specific bind both isoforms. As our data indicate that LC3A is capable of being recruited to stalled AVs, we would need a stringently LC3B-specific antibody to perform this experiment properly.

5) In Figure 6 the authors ask if WIPI2 could be involved in the stalling phenotype of AVs seen in neurons from aged mice. They should first show if WIPI2 is recruited or not to stalled AVs, as they have done for all other autophagy markers. Now, they directly shown the importance of WIPI2 and its binding to PI3P and ATG16L1 in autophagosome biogenesis in primary neurons, which is nice and expected, but the transition would have been better if they could show 6E-H before A-D (at least for WT WIPI2).

We appreciate the reviewer’s advice, and we have wrestled with the presentation of this data, but we feel that it is most efficiently presented as is. We did include immunocytochemistry of endogenous WIPI2 in fixed DRG neurons from aged mice (Figure 5—figure supplement 1F), showing that WIPI2 is recruited to stalled AVs.

6) In Figure 7 they show nicely that de-phosphorylation of WIPI2B at forming AVs seems to be required for recruitment of LC3B and AV biogenesis. However, they do not detect a difference in protein levels of total or phosphorylated WIPI2B with age and therefore propose that changes in cytosol/membrane levels of phosphorylated WIPI2B could change with age. This can be easily tested (and should be done) by a crude cytosol/membrane fractionation of their cell lysates or alternatively by gradient fractionation to see if the l specificity of membrane binding of P-WIPI2B is changed.

This was a great suggestion. As shown by the new data in Figure 8A-8B, we find that P-WIPI2 is significantly enriched in the cytosolic fraction of crude brain lysates in comparison to total WIPI2 levels. We see no change with age, as phospho-WIPI2 is more likely to be cytosolic than the total pool of WIPI2 in brain lysates from either young adult (3-month old) or aged (16-month old) mice. This observation fits with our hypothesis that the stalling observed in autophagosome biogenesis is a local event, due to local misregulation of WIPI2 phosphorylation. Our results regarding the residence time of the S395A versus S395E constructs also support this hypothesis.

Based on this experiment, we can extend our model to suggest that the recruitment of WIPI2B that is dephosphorylated at residue S395 is required to initiate autophagosome biogenesis (Figure 7D,E), but the local phosphorylation of WIPI2B at S395 is required to induce WIPI2B dissociation from membrane, concomitant with the growth of the nascent autophagosome (Figure 7F,G).

From the blots in Figure S4 and S5 it seems like the majority of WIPI2B is phosphorylated and that the upper band of the WIPI2 blot correspond to the P-WIPI2 band. Is this correct?

The endogenous mouse proteins from brain, P-WIPI2 and WIPI2, co-migrate on SDS-PAGE. In HeLa cells overexpressing the constructs, we can see the phosphomimetic migrates slightly slower than the nonphosphorylatable construct. We ran a Phostag (Wako) western blot to determine the relative levels of phosphorylated and unphosphorylated WIPI2 in mouse brain. Phostag gels restrict migration of phosphorylated proteins, allowing distinction between the forms using a single antibody. Our data suggest that approximately half of the total WIPI2 in phosphorylated in mouse brain across ages.

**Author response image 6. respfig6:** 

7) Figure 7H; this figure is confusing. The authors claim that it shows a stalled AV where WIPI2B S395E is not recruited, but how do we know ATG5 and WIPI2B S395E are expressed?

We now state more clearly in both the main text and the legend that these images are taken from neurons expressing ATG5 and WIPI2B S395E. We also include the time lapse sequence of the whole distal axon in Figure 7—figure supplement 2 to clearly show that this neuron is expressing each construct.

8) Figure 8B: the figure suggests a model where WIPI2B de-phosphorylation regulates its membrane recruitment and binding PI3P. To be able to conclude about this, they should do the membrane fractionation experiments suggested above and/or colocalization with PI3P.

As noted above, this was a great suggestion. The resulting data in the new Figure 8A-B, which indicate that phosphorylation of WIPI2 correlates with a decreased association with the membrane fraction from a brain lysate, further support the model in Figure 9 of the revised manuscript.

9) Furthermore, the model indicates that WIPI2B de-phosphorylation is important for the initial membrane recruitment of LC3B (and AV biogenesis), while WIPI2B re-phosphorylation is required for growth of LC3B positive structures. Their data are however not of sufficient strength to conclude about this and they should consider to modify the model as well as the sentence stating: " Further, we find that the dynamic regulation of WIPI2B phosphorylation at the isolation membrane is integral to autophagosome biogenesis, as our results suggest that dephosphorylated WIPI2B is required for recruitment of LC3B to the isolation membrane, while phosphorylated WIPI2B promotes expansion of the autophagosome".

We have revised the sentence as suggested, to read “Further, we propose that the dynamic regulation of WIPI2B phosphorylation at the isolation membrane is integral to autophagosome biogenesis. Our results indicate that only the nonphosphorylatable S395A form of WIPI2B is sufficient to rescue depletion of endogenous WIPI2, while recruitment of the phosphomimetic WIPI2B S395E correlates with expansion of the autophagosome.”

10) It is puzzling that the WIPI2B S395E mutant is not recruited to the early structures, as there is no difference in recruitment of DFCP1 (showing the presence of PI3P) and ATG5-12-16L1. Is the WIPI2B S395E mutant able to interact with PI3P and ATG16L1?

Both we (Figure 7—figure supplement 1B) and others (Wan, et al., 2018 Mol Cell) have shown that the WIPI2B S395E mutant still interacts with PI3P and ATG16L1. However, our new results indicate that phospho-WIPI2B has a lower affinity for the membrane fraction, suggesting that the affinity of one or more of these interactions is altered by phosphorylation. We plan to follow up on this interesting observation in future, in mechanistic studies using more tractable model systems.

Reviewer #3:The data are generally quantified and statistics included (with description of which test used in the figure legends), but the authors should include more information about the software used for quantification of rate of AV formation.

Rates of AV formation were calculated from changes in fluorescent intensities and area over time using FIJI (Schindelin et al., 2012). The quantification was done manually; it was not automated. We have clarified this point in the Materials and methods section of the revised manuscript.

This conclusion that autophagosome formation slows in aged mouse neurons is mainly supported by Figure 1 and Figure S1, which are quite convincing. Could the authors give more details about the definition of the rate of autophagosome biogenesis, i.e. LC3B puncta formation per min or mature autophagosome formation per min?

To quantify AV formation, we examined the GFP-LC3B channel in our time-lapse videos. A biogenesis event was defined as the de novo generation of a LC3B punctum or the growth of a punctum that was present at the start of the video acquisition (growth in terms of visible expansion of the area of the punctum). Since the latter category only applies to the puncta present at the start of the video, the vast majority of the events quantified were de novo GFP-LC3B puncta (that also grew during the time lapse). We did not quantify mature autophagosome formation. We have clarified this point in the Materials and methods section of the revised manuscript.

Figure 2, Figure 3, and part of Figure 5 support the conclusions about the step at which the defect occurs. Figure 4 and 5 show that Atg9 is retained in the stalled autophagosomes, convincingly. These findings are also quite clear. I am curious about the destiny of the stalled autophagosomes. Are they eventually degraded by lysosomes? This should be discussed.

This is a great question and something we want to follow up on in more detail. We were able to image stalled events for up to 20 minutes, and in that time the stalled structures diffused locally within the tip and did not recruit GFP-LC3B. We were not able to image longer than 20 minutes due to photobleaching of the fluorophore.

**Author response image 7. respfig7:** 

Figure 6 contains the key rescue data. It is interesting that WIPI2B function depends on Atg16-binding, but the Atg16 recruitment is not affected in aged mouse neuron. Please discuss this.

While ATG16L1 is indeed recruited to stalled AVs, we cannot be certain that its function is unaffected at these stalled AVs. We wanted to examine ATG16L1 dynamics in live-cell imaging, but overexpressing ATG16 appears to disrupt autophagosome biogenesis (data not shown), as seen previously (Li et al., 2017, Autophagy). Of note, the C-terminus of ATG16L1 can compensate for depletion of WIPI2 to maintain lipidation during starvation in cell lines (Lystad et al., 2019, Nat Cell Biol).

[Editors’ note: further revisions were requested prior to acceptance, as described below.]

Essential revisions:Although the authors have convincingly shown that overexpression of WIPI2B or a phospho-null version of this protein in aged neurons restores autophagosome biogenesis, the sequential phosphorylation model of WIPI2B to control autophagosome biogenesis should be more carefully discussed and main messages on dynamics toned down.

We now more clearly and thoughtfully discuss our observations on WIPI2B phosphorylation, and the working model on dynamic phosphorylation during autophagosome biogenesis that we are proposing. As requested, we toned down the language used when discussing the model to make it clear that this is a working model that needs further experimental validation.

1) Overexpression of both wild type or phospho-dead WIPI2(S395A) can rescue autophagosome biogenesis. How do the authors explain that a transition between dephosphorylation and phosphorylation is required for autophagosome biogenesis? The authors found that WIPI2 proteins levels increased with age in DRG neurons (Figure 6—figure supplement 1), but this result is discussed as if there is no change of WIPI2 levels with age. Combined with the observation that phosphorylation of WIPI2 is unchanged over time, this raises the possibility that overexpression of WIPI2 might not automatically lead to increases in phosphorylation (as argued by the authors in a rebuttal response). This brings about new questions, like how much (ie how many fold) is WIPI2 actually overexpressed in neurons compared to the age-associated increase observed, and has increases in WIPI2 phosphorylation (including over time) indeed been verified in this setting?

The only statistically significant change with age in WIPI2 protein levels was in DRG lysates from 1 mo-old mice and 24-mo-old mice. All other comparisons between ages were not significantly different. We do not conclude or model that phosphorylation of WIPI2 increases with age on a global or cellular level. In our rebuttal letter, we argued that overexpression of WIPI2 leads to increased nonphosphorylated WIPI2 available in the transfected neurons. If there was a typo in the rebuttal letter, we apologize for our mistake. Thus, we believe we agree with the reviewer on this point. We included immunoblot data of levels of WIPI2B overexpression in the first rebuttal letter that address the subsequent questions, but as noted we did not expect to see, nor did we see, increases in WIPI2 phosphorylation upon WIPI2B overexpression.

2) In Figure 5 A-E, the authors used co-localization of ATG9 with ATG5 to identify stalled AV. However, while the authors showed representative pictures in Figure 5—figure supplement 1 that some LC3B homologs co-localize with both ATG9 and ATG5, the authors used ATG5 alone to identify stalled AV in Figure 5 F-J. It is possible that these vesicles may be functional autophagic vesicles with other LC3 homologs instead of LC3B. The authors may consider these as stalled AV as ATG5 stay long on the vesicle, but an alternative explanation may be that autophagic vesicles with other LC3 homologs have a different dynamics compared to that of LC3B.

We used ATG5 colocalization with ATG9 to define stalled AVs in fixed neurons, as we could not observe temporal dynamics. When we used live-cell microscopy (as in Figure 5F-J), we used ATG5 residence time to define stalled AVs, as co-transfecting 3 constructs is technically difficult and would necessitate the use of a fourth laser line (405 nm), which only works in our hands with highly expressed constructs or markers labeling large organelles. These stalled AVs have other mAtg8s on them, as demonstrated by labeling mAtg8s in Figure 5F-J. However, the presence of other mAtg8s does not appear sufficient to reduce ATG5 residence time to under 5 minutes, which is what we have observed in young adult neurons (Figure 3). We agree that the other mAtg8s may have different dynamics than LC3B. However, since ATG5 residence time is not reduced (and thus, as defined, these AVs are stalled AVs), presence of other mAtg8s is not sufficient to explain the deficit we observed.

Figure 2L – The number of ATG5-positive punctae were found to decrease over time, as now directly stated in the paper, but the significance of this result is not discussed at all.

As indicated in Figure 2L, we did find a statistically significant decrease in the rate of formation of ATG5 puncta in DRG neurons from 24 mo old mice, suggesting that at this late time point additional aspects of autophagosome biogenesis may be affected. We now note this observation in the text.

3) Although the authors do not wish to make conclusions about their newly conducted flux assays, the provided diagram (which unfortunately is not accompanied with a figure legend, making it impossible to fully evaluate) may indicate that autophagy is still active in aged DRG neurons. If so, how would the authors explain the remaining autophagy activity with a decreased biogenesis of autophagosome vesicles in aged neurons?

A figure legend is now included below with the autophagic flux diagram initially provided in the first rebuttal letter.

As described in this work, we observed profound defects in autophagosome biogenesis, but do not see changes in bulk autophagy flux assays (see Author response image 2). This could be because the classic autophagic flux assay is not sensitive enough to detect the decrease in axonal autophagosome biogenesis that we observe. However, as we mention in the Discussion section, other aspects of the autophagy pathway may be altered with age, as suggested by Bejarano et al. Changes in later steps to autophagy would affect the autophagic flux assay downstream from autophagosome biogenesis, potentially masking the deficit we observed. We are looking at these steps in detail in follow-up work.

4) Missing data: Data showing ectopic overexpression of different mAtg8s to test LC3B recruitment to stalled AVs were not included, as requested by this reviewer – this makes it impossible to evaluate the full experiment, including the negative data.

We missed the reviewer’s point during the previous round of review. Figure 5—figure supplement 1, panel E, now includes data to address this point. Further, shown in Author response image 8 are western blot analysis of DRG cultures overexpressing the mAtg8s, demonstrating similar levels of expression.

**Author response image 8. respfig8:** DRG neurons were harvested from 16-17 mo GFP-LC3B transgenic mice and co-transfected with the indicated mCherry-mAtg8 construct and Halo-ATG5 to mimic the imaging experiments. Neurons were grown for 2 DIV, harvested, and immunoblotted for mCherry. Total protein for each lane was quantified: 100, 47, 83, and 57 for each lane, from left to right.

However, as we discussed in the first response to reviewers, this is a measure of the entire population of cultured DRG neurons, few of which are actually transfected. In our actual imaging experiments, we select neurons to image within a narrow fluorescence range. As fluorescence intensity directly correlates with protein expression, we can be certain that we are imaging and analyzing neurons expressing roughly similar levels of each of the mAtg8s. The Materials and methods were revised to further clarify this point.

Quantification data is missing in Figure 4E-F. Authors should provide how many events they have observed in young and old NMJs.

This quantification is now included in Figure 4G-I. The n values and relevant statistical information is provided in the Figure 4 figure legend.

5) The authors' argument not to cite Berjarano et al., 2018 because the study is not conducted in neurons is not valid. This study directly tested and showed, for the first time, an age-related decline in transport of autophagosomes, a conceptual advance that it relevant to this paper's findings, irrespective of cell type.

Bejarano et al., 2018 is now cited in the Discussion as requested.